



# Infiltration-Friendly Land Uses for Climate Resilience on Volcanic Slopes in the Rejoso Watershed, East Java, Indonesia

Didik Suprayogo[1], Widianto[1], Kurniatun Hairiah[1], Nabilla Meilasari[1], Abdul Lathief Rabbani[1], Rizky Maulana Ishaq[1], Meine van Noordwijk[1,2, 3]

[1]Soil Science Department, Faculty of Agriculture, Brawijaya University, Jl. Veteran no 1, Malang 65145, Indonesia
[2]World Agroforestry Centre, ICRAF, Indonesia Office, Bogor, Indonesia
[3]Plant Production Systems, Wageningen University, Wageningen, The Netherland

*Correspondence to*: Didik Suprayogo (Suprayogo@ub.ac.id and Suprayogo09@yahoo.com)

**Abstract.** Forest conversion to agriculture or agroforestry may increase risks of loss of hydrologic functions in an era of climate change. Infiltration during high-intensity rainfall is important for avoiding erosion and feeding aquifers but depends on land use practices that maintain soil macroporosity. In the forest-to-open-field-agriculture continuum it is not clear where thresholds to functionality ('degradation') are crossed. Our assessment of 'infiltration-friendly' land uses in the Rejoso

watershed on the slopes of the Bromo volcano in East Java (Indonesia) focused on two zones, upstream (above 800 m a.s.l.) and midstream (400-800 m a.s.l.) of the Rejoso river and feeding aquifers that support lowland rice areas as well as drinking water supplies to nearby cities. Upstream land uses included old and young pine plantations (production forest) and highland vegetable crops with variation of tree canopy cover. Midstream land uses included production forest, multistrata coffee-based agroforestry, clove-based agroforestry, and several mixed agroforestry types with variation of tree canopy cover. We quantified

infiltration and erosion in three replications per land use category over a three-month period (one-third of mean annual rainfall), with 6-13% of rainfall with intensities (51-100 mm day$^{-1}$). We related infiltration rates to plot-level characteristics across the land use systems and found statistically significant relations with tree canopy cover (likely based on combined effects of interception, preceding water use and effects on soil), understory cover, amount of litter, and soil surface roughness. Results for the upstream watershed showed that a tree canopy cover >55% is associated with adequate infiltration and acceptable soil

erosion levels. For vegetable cultivation in steep (45-65%) to very steep (> 65%) lands with a tree canopy cover below 55%, surface runoff was between 24% and 46% of rainfall, with high rates of soil erosion. Midstream, a tree canopy cover of >80% was associated with 'infiltration-friendly' land use, given the higher rainfall total (and rainfall intensity) in this zone. For a tree canopy cover in the range 20-80%, erosion rates were relatively low, but surface runoff increased to 36 to 62% of rainfall. Differences in soil type influenced the thresholds, as the areas' Inceptisols have lower intrinsic porosity than Andisols. A high

soil surface roughness and litter thickness assist in reducing surface runoff and soil erosion. Where more open forms of agroforestry, with tree canopy cover less than 80% are becoming more common this will affect water resources in the downstream area and increase vulnerability to climate change.





## 1. Introduction

Water access for all, Sustainable Development Goal 6 of the Agenda 2030 agreed by the United Nations (UN-Water, 2018),
not only refers to drinking-water and sanitation. It requires the protection of infiltration-friendly land uses in upland watersheds
as a source of clean water (Ostovar, 2019). Sufficient groundwater recharge is important to sustainable management of
groundwater resources to maintain streamflow throughout the year (Ali et al., 2016, Jacobs et al., 2018). While much of the
public discourse is in terms of forest versus agriculture, thresholds for specific soil and climate regimes are needed within the
intermediate agroforestry spectrum of land uses (van Noordwijk et al. 2014). Thresholds to critical hydrological functions are
likely dependent on local context but need to be understood to guide natural resource management in the challenging trade-
offs between local and external priorities (van Noordwijk et al. 2019). Wiersum (1984) documented that a high pine tree
canopy, without understory and permanent litter layer, can lead to high erosion rates due to high-impact drips from the needles.

Agroforestry systems with high canopy density can, if a permanent litter layer is present, maintain high infiltration
rates and can positively impact on hydrologic functions through: (1) a green canopy cover at tree and understorey level, (2)
land surface roughness, (3) litter at the soil surface, and (4) water uptake by trees and other vegetation (Hairiah et al., 2004,
Suprayogo et al., 2017). Five aspects that hydrologically differentiate natural forest from open-field agriculture, with
intermediate functionality for managed forest, plantations, agroforestry and trees outside forest (Creed and van Noordwijk,
2018; Jones et al. 2020) are: 1) the leaf area index (LAI) that allows photosynthesis when stomata are open and transpiring,
and that along with leaf morphology and rainfall intensity, determines canopy interception and subsequent evaporation, 2) the
surface litter that prevents crusting and supports infiltration (Liu et al., 2019), while reducing soil evaporation and reduces
entrainment of soil particles if overland flow still occurs, 3) the soil macroporosity that governs infiltration and allows for
aeration of deeper soil layers between rainfall events while recovering at decadal time scale after reforestation (Zhang et al.
2018, 2019), 4) the root systems that govern water extraction from deeper soil layers, in conjunction with the phenology of the
aboveground canopy, 5) possible influences on rainfall events (Ellison et al. 2017, 2019). Each of these five aspects has its
own dynamic (time constants) and dependency on the type of trees and their management, challenging definition of
hydrologically adequate land use choices. Rather than prescribing, independent of soil types and slope, the type and quantity
of tree cover that is needed as tends to happen in forest zoning, it may help if limits to infiltration-friendly land use (focussed
on the third function) can be operationalized in local context. In terms of watershed hydrology, infiltration-friendly land uses
can be interpreted as any land use that allows high rates of water infiltration so that surface runoff is a small (to be defined in
local context) fraction of rainfall and the watershed functions of flow buffering and erosion control are secured (to specified
standards). In watersheds that provide a perfect buffering river flow might theoretically be constant every day, but in practice
a 'flow persistence' metric of about 0.85 is hard to surpass (van Noordwijk et al., 2017). Flow buffering is essential for Climate
Resilience (Aduah et al., 2017; Shannon et al., 2019) and high flow persistence metrics are desirable, as they directly relate to
peak flow transmission (van Noordwijk et al., 2017).  .


On densely populated Java volcanic slopes are home to large numbers of farmers, while also serving as sources of water for lowland agriculture and the rapidly growing cities. The shrinking area of state-managed forests is no longer able to secure the required watershed functions, but at least part of the agroforestry managed by farmers can meet the required hydrological functions (Sajikumar and Remya 2015). In the Rejoso Watershed (Pasuruan, East Java, Indonesia), numerous stakeholders depend on the watershed functions of densely populated mountain slopes to meet their water demand. These include local

communities, farmers using water irrigation, the Regional Water Company, and bottled-water industries. A major infrastructure is planned to bring water to Surabaya and surrounding urban centres. However, the quantity and quality of the water at the source of the pipe have been decreasing over the past 10 years, putting the infrastructure investment at risk. Decreasing water resources are likely due to land use changes in the recharging area of Rejoso watershed on the northern slopes of the Bromo-Semeru volcanic mountain range, and/or decreased pressure on Artesian wells in the lands due to increased

extraction for paddy rice-fields. Among the hydrologic functions, infiltration is critical, as water travels through the subsoil to artesian wells at the foot of the volcano, in addition to surface rivers.

This research was thus designed to assess which land uses can maintain infiltration rates under local peak rainfall intensity and restrict soil erosion to acceptable levels. Specific questions were:

    1) Which existing land uses limit infiltration below the required rates at peak rainfall events?

2) Which factors that are directly observable such as tree cover, litter layer thickness or surface roughness can be used to define thresholds for 'infiltration-friendly land use'?

    3) Do answers to questions 1 and 2 need to be differentiated between the upper and middle watershed, with current vegetable production and agroforestry as dominant land uses, respectively?

## 2. Material and methods

### 85  2.1. Study area

The Rejoso watershed, is located in the foothills of Mount Bromo, covering 16 sub-districts in Pasuran District, East Java Province, Indonesia. The Rejoso watershed covers an area of 63,359 hectares with a watershed length of about 22 km. This study was conducted in two locations, namely in the upstream (above 800 m a.s.l.) and midstream (400-800 m a.s.l.) sections, with the dominant vegetation (land cover) selected for each location (Figure 1). In each location four dominant land use

systems were assessed (Table 1), spatially replicated in three separate measurement plots.

    ⇨ Fig. 1
    ⇨ Table 1



## 2.2 Rainfall

Rain gauges (ombrometers) were installed in four observation locations (with adjacent erosion plots) upstream and four
observation locations midstream of the Rejoso watershed. Each plot, the rainfall was measured with 5 replications. The
ombrometers consisted of a funnel and bottle with a volume of 1.5 dm$^3$ placed 120 cm above the soil surface and below the
tree canopy with bamboo as a support. Rainfall was observed every day during three months of the rainy season, from March
to May 2017.

## 2.3 Infiltration and soil erosion measurement

Infiltration was quantified in each land cover type via its complement, surface runoff, and expressed in the Runoff / Rainfall
ratio. As rainfall for quantified infiltration was measured below the tree canopy, the amounts were net of canopy retention.
Surface runoff was measured in 6 m x 2 m plots protected from surface run-on, with two drums at the lower end to collect
surface runoff and sediment concentrations for soil erosion measurement (Figure 2). In each plot, the water flow out through
13 holes of PVC pipes on drum-I was calibrated to be equal volume of water in each hole before runoff measurement. Runoff
samples were collected on every day at which rain occurred during the measurement period by measuring the water depth in
each drum. The amount of runoff in each rain even was calculated using eq. (1) and eq. (2):

$$R_t = V_{d-I} + (13 * V_{d-II}) \quad [1]$$

$$V_d = 10 * (\frac{\pi \, r^2 D}{A}) \qquad [2]$$

Where $R_t$ is total runoff (mm); $V_d$ is the water volume in drum I and II (mm), r is the radius of drum (cm), D is the water depth
in each drum (cm), and A is the areas of plot (cm$^2$).

The soil erosion in each rain even was determined by collecting 1000 cm$^3$ of runoff-sediment in each drum. The sample was
filtered with "newsprint" and dried in the oven with temperature 105°C to get the weight of sediment (S). The soil erosion in
each rain even was calculated using eq. (3):

$$E = (((\pi * r_{d-I}^2 * D_{d-I}) * S) + (13 * ((\pi * r_{d-I}^2 * D_{d-I})) * 10^{-6}) * S) * (\frac{10^8}{A}) \quad [3]$$

Where E is soil erosion (ton ha$^{-1}$); S is sediment (g).

⇨ Fig. 2





### 2.4 Determination of soil properties

Three bulk mineral soil samples were collected from the upper 20 cm below the litter layer at each site (A horizon). Particle
size distribution (particles <2 mm) was determined with the Bouyoucos densimeter method (Gee and Bauder, 1986) after $H_2O_2$
pre-treatment and after samples had been dispersed in a 5% sodium hexametaphosphate 5% dispersing solution. Bulk density
(oven dry weight per unit volume) was measured for a block-sized sample (20 cm x 20 cm x 10 cm = 4000 cm$^3$) collected at
field-moisture conditions (modified from Blake and Hartge, 1986). Total soil porosity (∅), the percentage of the total soil
volume that is not filled by solid (soil) particles (Nimmo, 2004), was calculated from bulk density data and particle density
using Eq. (4):

$$\emptyset = \left(1 - \frac{\rho_b}{\rho_p}\right) x \ 100\% \qquad [4]$$

where ∅ is porosity (%); $\rho_b$ is bulk density (g cm$^{-3}$), and $\rho_p$ is particle density (g cm$^{-3}$).

Soil organic carbon (SOC) was determined by dichromate oxidation (Walkley and Black, 1934).

### 2.5 Other plot characteristics

#### 2.5.1   Canopy cover

The percentage of canopy cover in each plot was calculated using the grid method, that records the shadow of sunshine at
ground level using plastic or paper (Arumsari, 2003 in Astutik, et al., 2015). The canopy projection when the sun was overhead
was drawn to scale on millimetre paper in each of four quadrants of the 20 m x 20 m plots, after which the areas shaded were
cut out and weighed separately. Canopy cover was calculated according to eq. (5):

$$\%CV = \frac{W \ Canopy}{W \ Total} X \ 100 \qquad [5]$$

Where: %CV is the percentage of tree canopy cover, W_Canopy is the paper weight representing canopy cover and W_Total
the paper weight representing the total area of observation, respectively.



### 2.5.2    Understorey and litter

Understorey vegetation and litter were measured according to the rapid carbon stock appraisal protocol (Hairiah et al., 2005), using 50 cm x 50 cm samples for fresh weight, with subsamples dried for dry weight determination.

### 2.5.3    Land surface roughness

Surface roughness was measured in each plot as the standard deviation of elevation measured every 30 cm along a thread (thin rope) installed 30 cm from the surface vertically, horizontally and diagonally over the erosion plot (Hoechstetter et al., 2008).

## 2.6. Data analysis

GenStat 15[th] edition was used for the statistical analysis of the results. The Fisher's LSD test, which establishes differences between groups defined for independent samples, was used for hypothesis testing, given that the data met the requirements for normality and homogeneity of variances. A probability level of 0.05 was set for rejecting null-hypotheses of no difference in tests of statistical significance. Linear regression relationships between the surface runoff / rainfall ratio or soil erosion and the

amount of rainfall, tree canopy cover, understory, litter, and land surface roughness were determined using SigmaPlot version10.0.

## 3    Results

### 3.1.    Rainfall

Within the measurement period 31 rainy days were recorded (Figure 3). Rainfall variation between upstream and midstream

observation plots was relatively high with an average of 520 mm (range 476 - 556 mm), and an average 666 mm (range 541 – 840 mm), respectively. In the upstream and midstream areas 71% and 57% of the rainy days had <20 mm day$^{-1}$ ('light rain'), 24% and 31% had 'moderate' rainfall (21-50 mm day$^{-1}$) and 6% and 13% 'heavy' rain (51-100 mm day$^{-1}$), respectively; none had 'very heavy rain' (> 100 mm day$^{-1}$). Such rain conditions indicate that the rain-erosivity in midstream is higher than that upstream.

⇨    Fig. 3

### 3.2.    Soil properties

The upstream area had lower bulk density and higher soil porosity, with lower clay content than the midstream area (Table 2). The C$_{org}$ concentrations varied from 0.65 to 2.12 %.

⇨    Table 2





## 3.3. Land characteristics related to runoff and soil erosion

Production forests in the upstream area had a lower tree canopy cover than those midstream but higher than those in agroforestry systems (Table 3). Agroforestry in the upstream area had a very low tree canopy cover because trees were planted only along field edges. Midstream agroforestry gardens ranged from high (75%) to low (26%) canopy cover. Understorey vegetation was more prominent upstream than midstream. Litter layer necromass and land surface roughness were generally aligned with tree canopy cover.

⇨   Table 3

## 3.4. Runoff and soil erosion

Decreasing tree canopy cover in agroforestry systems significantly increased the surface runoff / rainfall ratio (Table 4).

⇨   Table 4

Upstream production forests, with relatively coarse-textured soil (Table 2) had, despite steeper slopes (38-56%), a lower surface runoff / rainfall ratio than those in the midstream area with finer-textured soils and low land slope (3 -8%). In the upstream area, with decreasing tree canopy cover, the surface runoff / rainfall ratio increased 16-fold times compared to production forest (Figure 4.a). In the midstream area, agroforestry systems with a tree canopy cover > 80%, were still able to support low surface runoff (Figure 4.b). With a tree canopy cover of <80%, surface runoff increased rapidly on days with moderate rainfall (20-50 mm day$^{-1}$) (Figure 4.b.).

⇨   Fig. 4

In production forests with closed tree canopy cover, soil erosion rates were low (Table 4 and Figure 5 a.1, a.2 and b.1). These production forests still had a protective understorey vegetation, that contributed to litter necromass and surface roughness (Table 3), controlling splash erosion. Upstream, with reduction of tree cover canopy soil erosion increased dramatically from 20 to 110 times the rates measured in forested plots (Table 4).

⇨   Fig. 5

Erosion rates in all plots increased with rainfall intensity (Figure 5.a.3. and Figure 5.a.4). Midstream agroforestry systems had erosion rates range from 2.8 to 10.3 t ha$^{-1}$ in the measurement period (Table 4). As annual rainfall is approximately three times what was recorded in the measurement period, with similar rainfall intensities, these erosion rate are to be multiplied by a factor of 3, leading to 9 – 31 t ha$^{-1}$ year$^{-1}$. Even on volcanic soils, with frequent ash inputs, such erosion levels may be challenging sustainability.



### 3.5. Thresholds for infiltration-friendly land use

Increasing tree canopy cover, while maintaining understorey vegetation and litter necromass, is a strong indicator of watershed health as the main driver of low surface runoff (or high soil infiltration) and low soil erosion in production and agroforestry forest systems in the Rejoso watershed (Figure 6.a.1, 6.b.1. and Figure 7.a.1, 7.b.1).

⇨ Fig. 6

Understorey vegetation reduces splash impacts on the soil and supports infiltration, as does the litter necromass present. (Figures 6.a.3, 6.b.3. and Figure 7.a.3, 7.b.3). Land surface roughness, in contrast to litter necromass, had no consistent relationship with runoff or erosion (Figure 6.a.4. and Figure 7.a.4).

⇨ Fig. 7

## 4. Discussion

A number of land cover types had infiltration rates below the required rates at peak rainfall events. Among the four factors tested, tree cover and litter layer necromass could be used to define zone-specific thresholds for infiltration-friendly land use, but understory vegetation and surface roughness did not. Although slopes in the upper watershed are much steeper than midstream, the coarser texture and likely higher aggregate stability means that thresholds for canopy cover and litter necromass can be lower.

Many authors have emphasized that the key to hydrologic functions is in the soil rather than the aboveground parts of the forest (Peña-Arancibia et al. 2019). Still, we found strong and direct relations with canopy cover. Positive effects of canopy cover on infiltration were related to raindrop interception in earlier studies (Carlesso et al. 2011; de Almeida et al. 2018). Interception will (a) reduce the destructive power of rainwater splash on the ground surface (as long as the effects Wiersum (1974) described are avoided, (b) allow more time for infiltration as water reaches the surface more slowly, (c) keep a thin water film on the leaves that will (d) cool the surrounding air when it subsequently evaporates. It will reduce the amount of water reaching the soil surface, but by increasing air humidity also decrease transpiration demand when stomata are open. In a study in North China, Li et al. (2014) showed that presence of litter of *Quercus variabilis*, representing broadleaf litter, and *Pinus tabulaeformis*, representing needle leaf litter, can reduce surface runoff rates by 29.5% and 31.3% respectively. The overall effect of fast plus slowly decomposing surface litter means protection of the soil surface from splash erosion, surface roughness that reduces sediment entrainment, an energy source for soil biota and a conducive microclimate (Hairiah et al. 2006, Derpsch et al., 2014).

Both the production forest and agroforestry systems maintained a relatively high land surface roughness. Without a high canopy cover this roughness was not able to control surface runoff and erosion in the upstream area, but midstream there were significant correlations. The role of surface roughness as sediment filter may depend on frequent regeneration to counter





homogenisation (Rodenburg et al. 2003). The soils' macroporosity, needed for effective infiltration, is the result of a continuous process of compaction and filling in of macropores with fine soil particles, and creation of biogenic channels (formed by old tree roots, earthworms and other soil engineers) or abiotic processes (cracks). As no heavy machinery is used in any of these

land use systems, compaction is restricted to human feet, and motorbikes in specific tracks. The formation of old tree root channels can cause long time-lags between land cover change and soil macroporosity (van Noordwijk et al. 2011), obscuring relations between current tree cover and hydrologic functions. Zwartendijk et al. (2017) showed that 'fallows' were intermediate between forests and grasslands in terms of infiltration in Madagascar. Recovery of infiltration after reforestation of grasslands in the Philippines was found to be a matter of decades rather than years (Zhang et al. 2019).

From a land use policy perspective our results suggest that maintaining high (~80%) canopy cover in the mid-slope farmer-controlled landscape that does not match the slope criteria for designation as watershed protection forest, is important. In Indonesia, protection is forest areas that have the primary function as protection of life support systems to regulate water management, prevent flooding, control soil erosion, prevent sea water intrusion, and maintain soil fertility (Government of Indonesia, 1999). With the higher rainfall intensities here and more erosive soils, risks for degradation from a downstream

perspective seem to be as important here as they are in the more visually-at-risk upper watershed zone. Combining our plot-level results with efforts for hydrologic modelling for the Rejoso catchment as a whole (Tanika et al. 2018) can guide further advice to a local watershed forum on the measures and incentives needed to restore and protect the watershed as a whole. The Indonesian legal requirement of 30% forest cover across all its local government entities (Government of Indonesia, 2007) is a coarse translation of hydrologic relations at risk. It clearly matters what the land cover in the 'non-forest' parts of the

landscape is and how vegetation interacts with soils and geomorphology in shaping rivers and groundwater flows (Zhipeng et al. 2018; Zhao et al. 2019). Our findings for the Rejoso watershed show that within the agroforestry spectrum, hydrologic thresholds of infiltration-friendliness exist between the systems that are mostly 'agro' and those that are mostly 'forest'.

## 5.    Conclusions

Our results demonstrated that vegetation-based thresholds for adequacy of infiltration, given the existing rainfall intensities,

differed between the upper and middle watershed. Despite steep slopes and low tree cover, the upper watershed with its coarse soil texture and strong aggregation typical of Andosols, land management practices that combine vegetable crops with a tree canopy cover of around 55% can maintain infiltration and reduce erosion to acceptable levels. In the midstream part of the catchment, despite gentle slopes, infiltration-friendly land use on the fine-textured Inceptisols required a canopy cover of 80%. Beyond tree canopy cover, litter layer necromass was found to be a good and easily observed indicator of infiltration rates,

while understory vegetation and surface roughness may support infiltration, but as such are not sufficiently strong indicators.



**Author contributions**. DS, W, KH and MvN was designed the study. NM was collected data in midstream, ALR was collected data in upstream, RMI was coordinated to collect the data in the field, academically supervised by DS, W and KH. Didik Suprayogo, DS, W, KH and MvN shaped the manuscript, which was approved by all co-authors.

**Competing interests**. The authors declare that they have no conflict of interest.

**Acknowledgements**

Authors thank to the community of Rejoso watershed and the "*Rejoso Kita*" Forum. They also thank the Department of Soil Science, Faculty of Agriculture, University of Brawijaya and the Research Group of Tropical Agroforestry for support of the research. Finally, the authors would like to thank the Social Investment Indonesia (SII) organisation for connecting the local stakeholders during field work.

**Financial support**. This research has been supported by the Danone Ecosystem Fund via the World Agroforestry Centre, ICRAF, Indonesia Office. This research is **also** partially funded by the Indonesian Ministry of Research, Technology and Higher Education under WCU Program managed by Institute of Research and Community Services Universitas Brawijaya and Institut Teknologi Bandung.

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





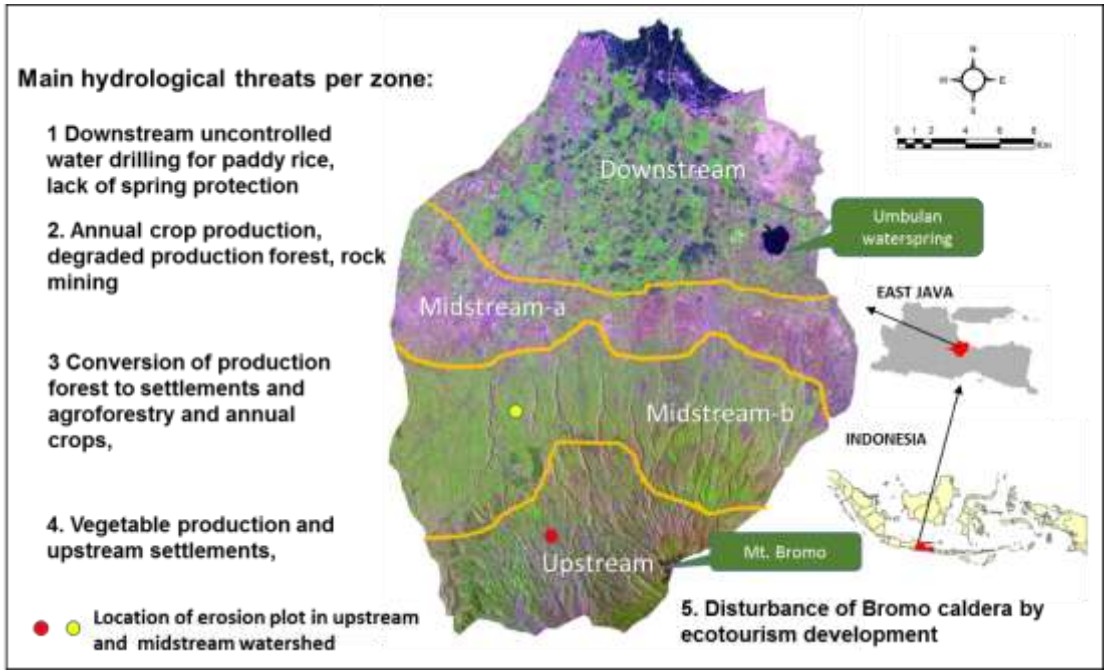


Figure 1. The Rejoso Watershed from upstream (at the bottom) to sea level and land uses considered to be a hydrological threat; purple indicates open soil, green tree cover. (Modified from USGS, 2019).

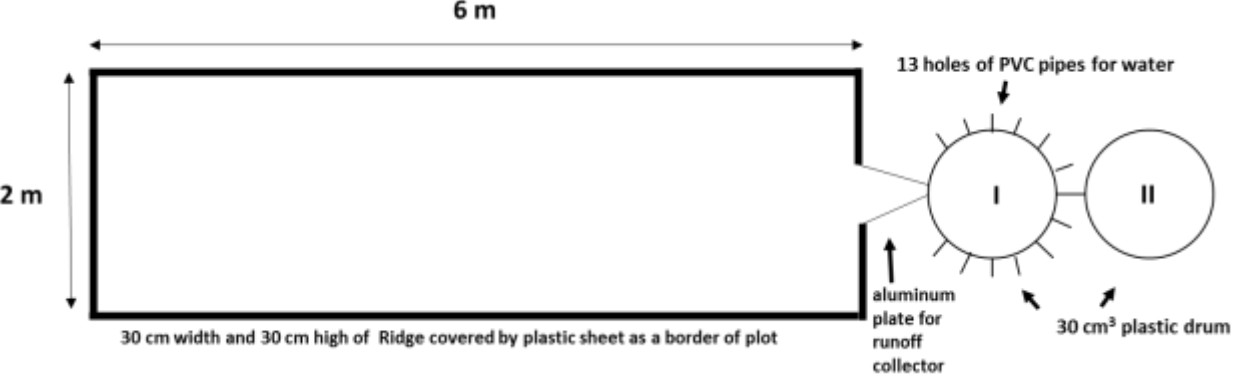

Figure 2. Runoff and soil erosion plot design.





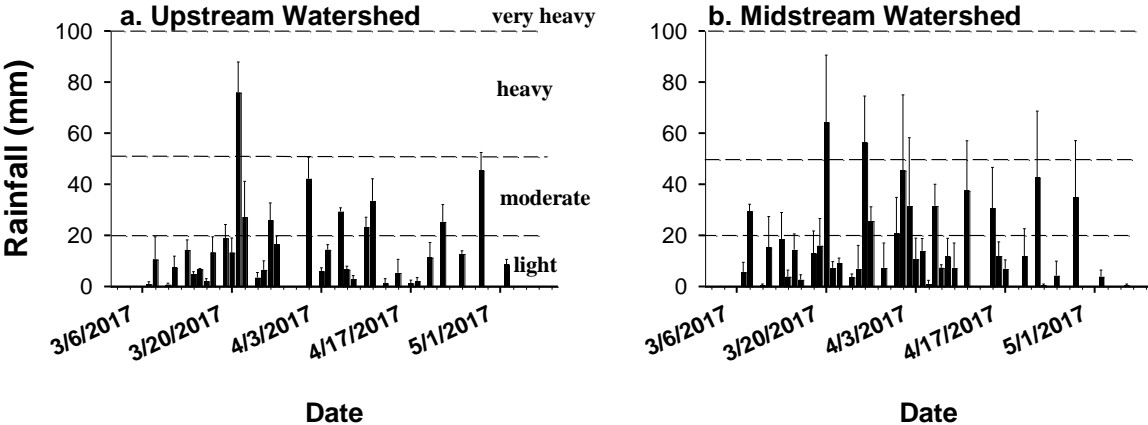


Figure 3. Distribution of rainfall during observation between March to May 2017 in the Rejoso watershed.

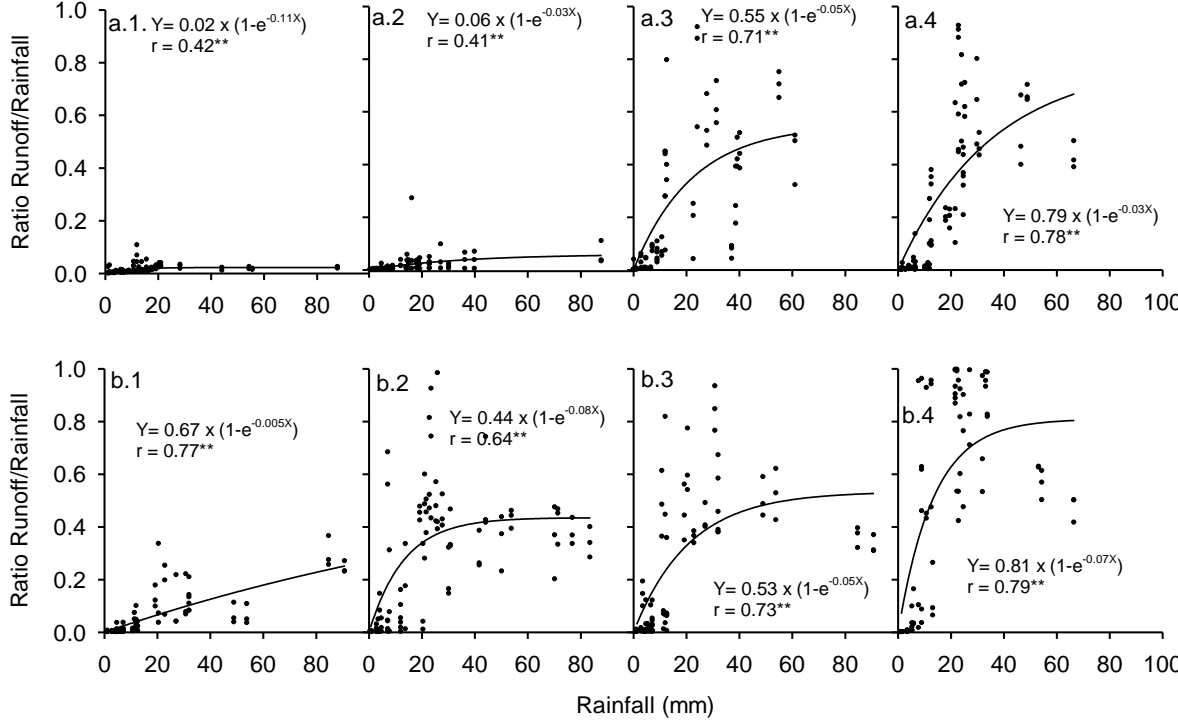

Figure 4. The relationship between surface runoff / rainfall ratio and the amount of rainfall in production forest and

agroforestry systems in the Rejoso watershed (a: Upstream Rejoso Watershed; b: Midstream Rejoso Watershed).





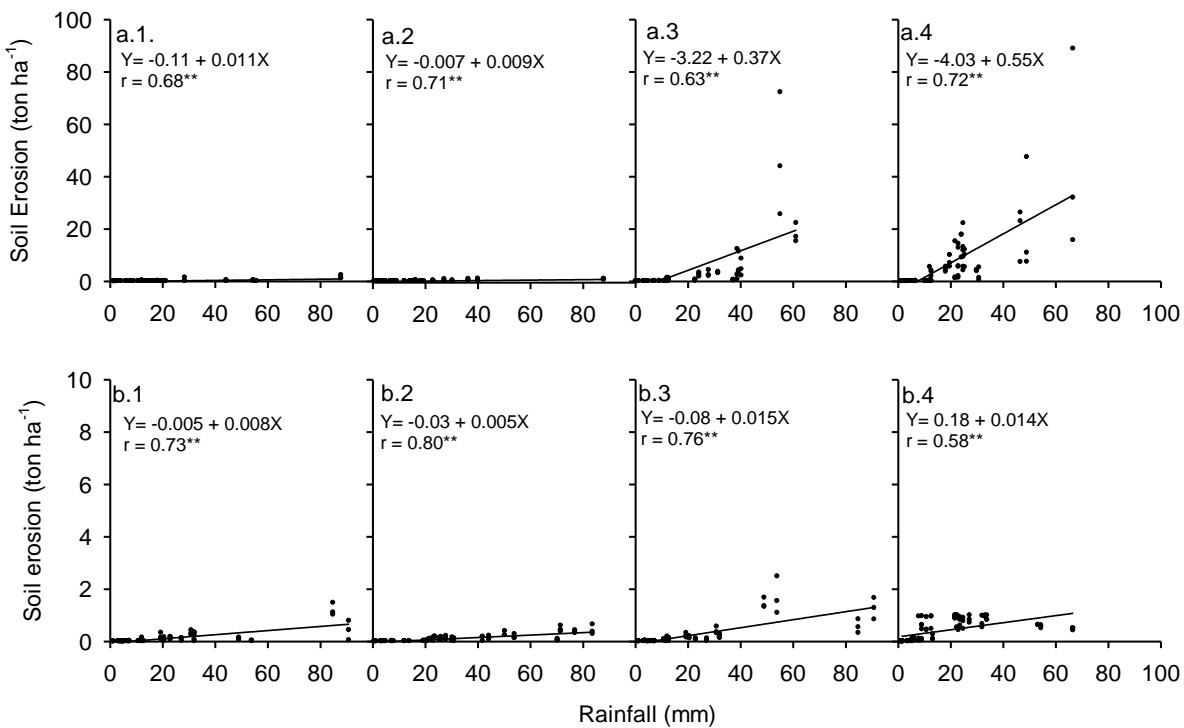


Figure 5: Soil erosion in relation to daily rainfall rates in production forest and agroforestry (a: Upstream Rejoso Watershed; b: Midstream Rejoso Watershed)


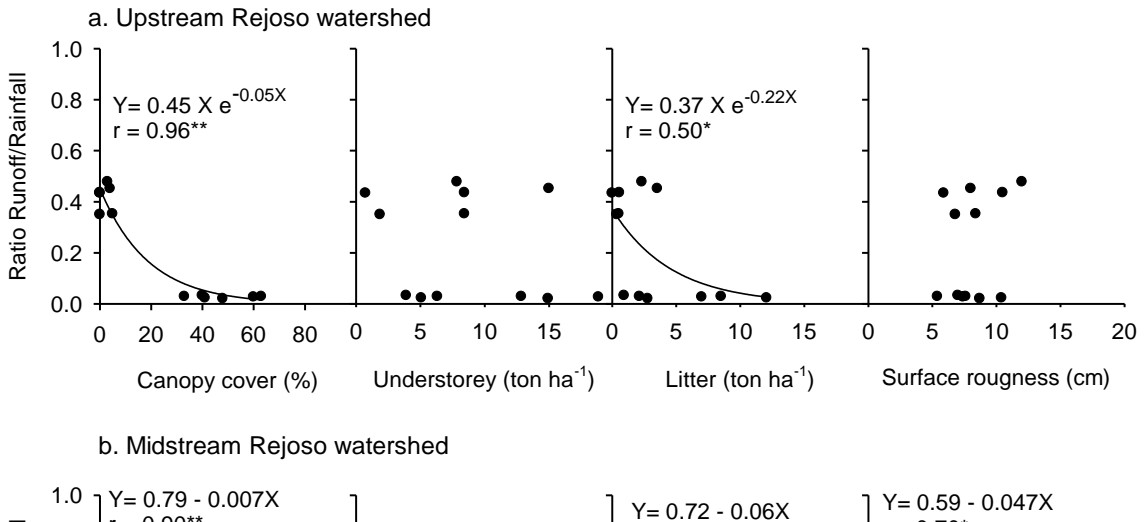

Figure 6. The runoff / rainfall ratio as function of tree canopy cover, understorey vegetation, litter necromass, and land surface roughness.



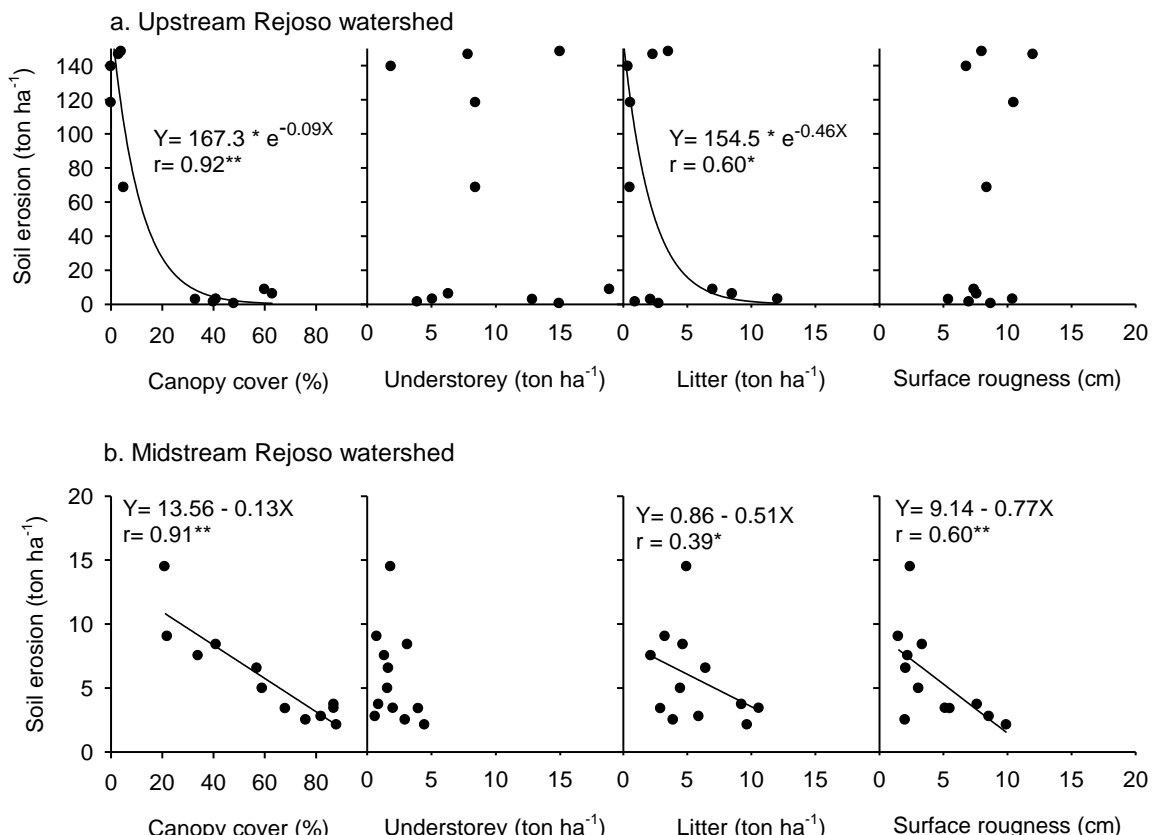

Figure 7. Soil erosion in relation to tree canopy cover, understorey vegetation, litter necromass, and land surface roughness.






Table 1. Land use, vegetation, soil conservation measure and slope of measurement plots

| Code | Land use | Vegetation | Terracing | Slope at plot level (%) |
|---|---|---|---|---|
| Upstream Rejoso watershed | | | | |
| UT1 | Old production forest | Pine (*Pinus merkusii*) + grass | None | 35-40 |
| UT2 | Young production forest | Pine + grass | None | 50-60 |
| UT3 | Agroforestry | Strip cemara (*Casuarina junghuniana*) + Cabbage | None | 40-50 |
| UT4 | Arable land | Banana, maize, carrot | None | 40-50 |
| Midstream Rejoso watershed | | | | |
| MT1 | Old production forest | Mix Pine or mahoni (*Swietenia macrophylla*), banana, salak (*Salacca zalacca*), taro (*Colocasia esculenta*), elephant grass (*Miscanthus giganteus*). | Bench terrace | 3-8 |
| MT2 | Agroforestry | Coffee-based mix with durian (*Durio zibethinus*), mahoni, *Leucaena leucocaphala*, *Paraserianthes falcataria*, *Albizia saman*, dadap (*Erythrina variegata*), banana | Bench terrace | 3-8 |
| MT3 | Agroforestry | Clove (*Syzygium aromaticum*), banana | Bench terrace | 3-8 |
| MT4 | Agroforestry | Manggo (*Mangifera indica*), durian, dapap, maize, cassava, groundnut | Bench terrace | 3-8 |






Table 2. Soil texture, bulk density, soil porosity and organic C of runoff plots

| Location Code | Soil texture (%)* | | | Bulk Density (g cm⁻³)* | Particle Density (g cm⁻³)* | Soil porosity (%)* | C$_{org}$ (%)* |
|---|---|---|---|---|---|---|---|
| | Sand | Silt | Clay | | | | |
| Upstream Rejoso watershed: Andisols | | | | | | | |
| UT1 | 18 a | 52 ab | 31 a | 0.78 a | 2.20 a | 65 a | 2.45 b |
| UT2 | 17 a | 49 a | 34 a | 0.79 a | 2.36 b | 67 a | 1.70 a |
| UT3 | 12 a | 55 b | 33 a | 0.81 ab | 2.20 a | 63 a | 2.00 ab |
| UT4 | 14 a | 54 ab | 33 a | 0.95 b | 2.43 b | 61 a | 1.47 b |
| LSD | | 4.86 | | 0.14 | 0.14 | 6.9 | 0.66 |
| Midstream Rejoso watershed: Inceptisols | | | | | | | |
| MT1 | 11 a | 48 b | 41 a | 0.88 a | 2.26 a | 61 a | 1.58 b |
| MT2 | 10 a | 43 ab | 47ab | 0.93 ab | 2.34 a | 62 a | 2.02 b |
| MT3 | 7 a | 46 b | 47 ab | 1.08 b | 2.42 a | 56 b | 1.28 ab |
| MT4 | 11 a | 35 a | 54 b | 1.07 b | 2.35 a | 55 b | 0.65 a |
| LSD | | 9.0 | 11.7 | 0.15 | | 3.44 | 0.78 |

*The same letter indicates no statistically significant differences between location with Fisher's LSD test (p<0.05).








Table 3. Canopy cover, understory vegetation, litter necromass, and soil roughness of the sample plots

| Code | Land cover | Tree canopy cover (%)* | Understorey vegetation (t ha⁻¹)* | Litter (t ha⁻¹)* | Soil roughness (%)* |
|------|-----------|------------------------|-----------------------------------|------------------|----------------------|
| | | | Upstream Rejoso watershed | | |
| UT1 | Old production forest | 55 b | 10.1 b | 9.2 b | 8.5 a |
| UT2 | Young pro-duction forest | 40 b | 10.5 b | 2.0 a | 7.0 a |
| UT3 | Agroforestry | 4 a | 10.1 b | 2.1 a | 9.5 a |
| UT4 | Arable land | 0 a | 3.7 a | 0.3 a | 7.7 a |
| LSD | | 15 | 5.6 | 3.7 | 4.6 |
| | | | Midstream Rejoso watershed | | |
| MT1 | Old production forest | 87 c | 2.5 a | 9.8 b | 7.6 b |
| MT2 | Agroforestry | 75 c | 2.5 a | 4.8 a | 5.4 ab |
| MT3 | Agroforestry | 52 b | 2.1 a | 5.2 a | 2.8 a |
| MT4 | Agroforestry | 26 a | 1.3 a | 3.5 a | 2.0 a |
| LSD | | 14 | 2.6 | 2.4 | 4.5 |

*The same letter indicates no statistically significant differences between location with Fisher's LSD test ($p<0.05$).





Table 4. Rainfall, runoff, ratio runoff/rainfall and soil erosion in the runoff plots in each land cover type

| Code | Land cover | Ranfall (mm) | Runoff (mm)* | Runoff/ rainfall ratio* | Soil erosion (ton ha⁻¹)* |
|---|---|---|---|---|---|
| Upstream Rejoso watershed | | | | | |
| UT1 | Old production forest | 555 | 14.3 a | 0.03 a | 5.86 a |
| UT2 | Young production forest | 492 | 13.2 a | 0.03 a | 1.47 a |
| UT3 | Agroforestry | 476 | 203.3 b | 0.43 b | 120.98 b |
| UT4 | Arable land | 556 | 225.7 b | 0.41 b | 163.22 b |
| LSD | | | 46.3 | 0.09 | 87 |
| Midstream Rejoso watershed | | | | | |
| MT1 | Old production forest | 616 | 80.2 a | 0.13 a | 3.07 a |
| MT2 | Agroforestry | 841 | 316.3 c | 0.38 b | 2.88 a |
| MT3 | Agroforestry | 616 | 228.8 b | 0.37 b | 6.63 ab |
| MT4 | Agroforestry | 541 | 344.9 c | 0.64 c | 10.33 b |
| LSD | | | 86.6 | 0.12 | 4.22 |

*The same letter indicates no statistically significant differences between location with Fisher's LSD test (p<0.05).