# Peer review of "Infiltration-Friendly Land Uses for Climate Resilience on Volcanic Slopes in the Rejoso Watershed, East Java, Indonesia"

_Hydrology and Earth System Sciences, 2020_

## Referee Comment (RC1) · Sampurno Bruijnzeel (Referee) · 15 Mar 2020

Comment on 'Infiltration-friendly land uses for climate resilience on volcanic slopes in the Rejoso watershed, East Java, Indonesia' by D. Suprayogo et al. (HESSD; https://doi.org/10.5194/hess-2020-2)

L. Adrian Bruijnzeel, Department of Geography, King's College London, Strand, London WC2 2LS, U.K. E-mail: sampurno.bruijnzeel@kcl.ac.uk

General comment: the paper by Suprayogo et al. describes the results of a short-term (ca. two months) field experiment aiming to identify which of the dominant land uses

in the middle and upper parts of the 633 km2 upland Rejoso catchment in the volcanic uplands of East Java may be considered 'infiltration-friendly' under the prevailing rainfall intensities. In addition, the study seeks to identify which readily measured vegetation and surface characteristics may be used to define the critical threshold values associated with such 'infiltration-friendly' land uses. The rationale for this work partly lay in an observed decline in the quantity and quality of water resources in the study area believed to reflect changes in land use and land cover in the Rejoso basin as well as increased extraction of groundwater for irrigated rice cultivation in the lower parts of the catchment. The authors rightly point out that much of the debate on land cover and catchment water yield and/or streamflow regimes focuses on forested versus agricultural uses of the land, whereas preciously little is known on the comparative ability of such 'intermediate' land-use types as agroforestry systems in terms of maintaining soil infiltration capacity, groundwater recharge and dry-season flows under seasonal tropical conditions (Toohey et al., 2018; Nespoulos et al., 2019). Likewise, most global reviews of the literature on 'forests and water' have concentrated on the changes in total water yield associated with forest removal or addition, interpreting the observed changes in flow primarily in terms of increases or decreases in vegetation water use (evapotranspiration, ET) as a function of forest type or climate, but leaving the effects on flow regime by potential changes in surface conditions essentially non-analyzed (e.g. Zhang et al., 2017; Filoso et al., 2017; Bentley & Coomes, 2019). Only a few studies have paid explicit attention to changes in seasonal streamflow regime after removing or adding trees (e.g. Lane et al., 2005; Brown et al., 2013; cf. Van Noordwijk et al., 2017ab). In view of the fact that such changes in streamflow regime will reflect both changes in ET and in infiltration after the change in land use under given climatic conditions (Bruijnzeel, 2004; Peña-Arancíbia et al., 2019) the paper by Suprayogo et al. is to be welcomed in principle in that it documents infiltration (at the runoff plot scale) and erosion rates for a series of agroforestry systems (both terraced and non-terraced), rain-fed crops (mostly maize) and forest plantations (pine and mahogany) that may be considered typical for Java's densely populated uplands. Moreover, the paper marks

a valiant attempt at identifying some of the key vegetation and surface characteristics controlling infiltration rates, runoff production (i.e. overland flow) and surface erosion rates using low-cost / low-tech approaches for measuring rainfall, runoff and site characteristics.

The paper's key findings include strongly negative relationships between canopy cover fraction and either plot-scale runoff coefficient (Fig. 6) or surface erosion (Fig. 7), and to a lesser extent between the latter and surface litter amounts while understory biomass seems less important. However, individual land-use types deviated from this general pattern. For example, the 'young' and 'old' production forest plots (UT1 and UT2) had the same runoff coefficients (13-14%) but the young forest exhibited a far smaller soil loss than the older forest, despite having a more open canopy (by 12%, difference not significant), a much lower litter mass (2.0 vs. 9.2 t ha-1) and a much steeper slope (50-60% vs. 30-40%) (Tables 3 and 1). Likewise, agroforestry plots MT2 and MT3 exhibited the same (high) runoff coefficient (37-38%) but differed by a factor of 2.3 in soil loss with the amount of litter on the ground being the same (Tables 3 and 1). Such findings suggest that soil characteristics (as opposed to surface or vegetation variables) likely play a role as well and perhaps deserve to be included in any predictive equations (e.g. SOC?; applicable in the case of MT2 vs. MT3 but not for UT1 vs. UT2 (Table 2)). At any rate, it would be good to include such considerations explicitly in the Discussion section. On the downside, the paper gives the distinct impression of having been put together in some haste and the often highly concise text leaves much to be guessed (or derived) by the reader. This holds especially true for the sections describing the study area and methods, but also for the Results and Discussion. For example, the study area description effectively consists of a Table describing the land-cover types for the eight examined locations (Table 1) but fails to give basic location or climatic information, or a proper description of vegetation characteristics (e.g. tree height – important to assess the erosive power of crown drip) or the nature of the terraces of the mid-stream research plots (notably whether the plots included terrace risers or terrace beds only). As to the methods used, it is not clear – inter alia – in

which plots 'rainfall' truly represents incident rainfall or rather crown drip (throughfall) (line 101); what size or type of funnel was used for the improvised rain gauges and what the catch efficiency of these gauges was compared to that of a standard rain gauge (cf. Ghimire et al., 2017; Zhang et al., 2019); how the collection drums with a stated capacity of 30 cm3 (!) (Fig. 2) could accommodate the runoff produced by 12 m2 plots with runoff coefficients of up to 41-64% (Table 4); how coarse sediment eroded from the plots was accounted for or what the effect of filtering runoff samples through 'newsprint' (lines 111-112) was on the magnitude of sediment concentrations obtained compared to more conventional filtration methods; how particle density was determined (line 124); or how many replications were used to determine undergrowth or litter mass (lines 140-141), etc. Ibidem for the Results section: e.g. the main results for soil properties are described in 1.5 lines only (lines 162-163); key Tables 2-4 give averages or period totals only but no standard errors or coefficients of variation as a measure of within-plot variability despite the fact that the large variation in rainfall (throughfall?) totals between adjacent (?) plots (e.g. 300 mm for plots MT4 and MT2 in Table 4) suggested major spatial variability; the captions to key diagrams 4 and 5 which count 8 panels each do not explicitly specify which panel refers to which plot in any descriptive way other than their relative position in the list of plots in Table 1. The Discussion section is rather basic and does not address such critical issues as the improvised nature of the rain gauges and the neglecting of stemflow as a possible input of water to the soil (which would lead to under-estimation of 'rainfall' and hence over-estimation of runoff coefficients), as well as the scale and ability of the plot-based measurements to represent the entire hilllslope's hydrological functioning (see below for details and related issues)). Furthermore, the reference list – although remarkably up to date with more than 60% of cited references published between 2017 and 2019 – misses at least six references that are cited in the text and contains another six that are listed all right but do not show up in the text (see below under specific comments).

On a related note, rather than to refer largely to recent studies in such very different environments as the semi-arid loess plateau in China (Zhao et al., 2019; Liu et al.,

2018 – erroneously referred to as Zhipeng et al., 2018) the paper would arguably have benefitted from the inclusion of related studies in the volcanic uplands of West and East Java for added perspective. Examples include the effect of trampling/footpaths on runoff production and erosion from upland fields (Bons, 1990; Rijsdijk et al., 2007; cf. Badu et al., 2019); the role of terrace risers versus terrace beds vis á vis runoff and sediment production (Purwanto and Bruijnzeel, 1998; Van Dijk & Bruijnzeel, 2004ab); the potential role of stemflow in the generation of localized infiltration or overland flow (high stemflow fractions reported for bamboo, bananas, shaded coffee, tall grasses and understory shrubs (Taniguchi et al., 1996; Cattan et al., 2007; Siles et al., 2010; Friesen et al., 2013; González-Martínez et al. 2016)); as well as a more holistic discussion of the interactions between canopy cover fraction, understory, leaf litter and hydrological processes (Wiersum, 1985; cf. Prijono et al., 2014; even Coster, 1938) – not only in terms of amounts of water involved but also the erosive power of the rain / crown drip as a function of falling height and leaf type (Wiersum, 1985; Hall and Calder, 1993). In contrast to what is stated in the paper (lines 209-211), rainfall interception does not reduce the erosive power of the rain. Rather, a tree canopy enhances it because of the typically larger drop size of crown drip compared to incident rainfall for terminal fall velocities (Wiersum, 1985), with broad-leaved species producing larger drops than do conifers (Hall and Calder, 1993). Likewise, the few measurements of throughfall intensities under humid tropical conditions suggest these to be very similar to those of incident rainfall (e.g. Hutjes et al., 1990). As such, the presumed effect on delaying the onset of overland flow (line 211) would seem rather negligible. Similarly, based on the lack of correlations with surface runoff or erosion (Figs. 6 and 7) the role of understory vegetation is considered to be minor. Yet Nespoulos et al. (2019) emphasized the importance of a well-developed understory for the development of macroporosity and preferential pathways in tropical plantations. In addition, the discussion could use a paragraph on the importance of including both infiltration and total ET (not just interception loss) for a proper assessment of the effect of land cover on groundwater recharge.

Other points not considered in the Discussion include such aspects like (a) the need for an adequate number of throughfall gauges to quantify net precipitation beneath such spatially highly variable vegetation types as some of the studied agroforestry systems (cf. Holwerda et al., 2006; Zion Klos et al., 2014); (b) the role of auto-correlation in the presented relations between runoff coefficient (i.e. runoff/ rainfall) and rainfall, (c) the possible role of (high) short-term rainfall intensities as opposed to the presently used daily totals in determining measured amounts of surface runoff and eroded soil (cf. Wischmeier's EI30 index), (d) the effect of the small size of the runoff plots used (2 m x 6 m) on measured amounts of surface runoff vis á vis their representativeness for hydrological functioning at the hillslope scale (variations in slope steepness, re-infiltration on less steep foot-slopes in the case of the upper plots; cf. Stomph et al., 2002; Moreno-de las Heras et al., 2010), and (d) what might constituted a plausible value of 'tolerable soil loss' for the study area (Verheijen et al., 2009; cf. Bruijnzeel, 1983; see also specific comment *22 below for a possibly locally valid estimate).

On the basis of the above considerations my recommendation for this manuscript can only be rejection in its present form but allow resubmission if the main points raised in the previous paragraphs can be addressed satisfactorily. After all, the paper addresses an important topic and new information on infiltration behaviour of agroforestry systems is to be welcomed.

Specific comments: *1, Title: you may want to use inverted commas for 'infiltration-friendly' in the title as well. *2, line 20: based on the time line in Figure 3 (6 March–1 May 2017) a period of nearly two months would be more accurate than 'three months'. *3, line 21: strictly speaking, when using daily rainfall and surface runoff totals to obtain net infiltration amounts one cannot refer to the latter as infiltration rates. *4, line 24: 'preceding water use' is unnecessarily vague; suggest replacing by 'soil moisture status' because moisture values are governed by the interplay of rainfall/drip, evaporation and soil water uptake, not just vegetation water uptake. *5, line 24: relations with understory biomass or surface micro-topographic variation were not strong or absent

(Figs. 6 and 7). *6, line 28: an average runoff coefficient of 62% (actually 64% in Table 4) is exceedingly high and more representative of compacted dirt roads and yards in the area than actively worked agricultural fields (see e.g. Rijsdijk et al., 2007). Such high values might suggest rainfall inputs may well have been under-estimated. Worth checking! *7, line 29: with porosities of 55-62% and bulk densities of 0.9-1.1 g cm-3 (Table 2) the soils of the mid-stream sites are not particularly dense. Rather, one would think of crusting or slaking as a potential cause for low apparent infiltration? *8, lines 41-42: this sentence seems to fall out of the blue; since the cited reference concerns a global review of the literature on surface erosion it might be possible that the authors meant Wiersum (1985) instead which documents the role of understory and litter layer with regards to surface runoff and erosion in an Acacia auriculiformis plantation in West Java, not pines. Incidently, drip from a pine canopy is less erosive than that from broad-leaved species like Eucalyptus and, especially Teak (Hall and Calder, 1993). *9, line 52: the sentence on infiltration recovery seems out of place here and had perhaps better be moved to the Discussion section. *10, line 54: whilst the influence of a very extensive and aerodynamically rough forest cover on rainfall may have an effect on downwind rainfall amounts (as opposed to 'events'), it would seem inappropriate to mention this in the present context of on-site water dynamics. Suggest to leave out this aspect. *11, lines 62-64: unclear why Climate Resilience is written with initial capitals?; also, relation between flow persistence metrics and peakflow transmission (routing? percolation?) is not instantly clear. *12, lines 77-78: soil fertility and agricultural productivity may be maintained sufficiently on these deep volcanic deposits even if surface erosion rates are high. Also, previous research on sediment production in similar terrain nearby in East Java (Rijsdijk and Bruijnzeel, 1990ab; Rijsdijk, 2005) has shown that contributions from rain-fed agricultural fields made up a comparatively minor proportion of overall annual sediment yield at the operational catchment scale with roads, paths, settlements contributing significant amounts each. *13, section 2.1, Study area: suggest to give a proper basic description of site locations (place names, latitude, longitude, elevations) along with information on the main environmental conditions, notably (a) annual rainfall totals and the agro-climatic classification of the two main sites in terms of rainfall seasonality (e.g. Oldeman climate types D3 and C2 for the middle and upper zones of the catchment?), (b) prevailing rainfall intensities (e.g. based on the authors their own measurements or the iso-erodent map of Java ?, and perhaps (c) FAO reference evapotranspiration (to help assess the importance of differences in infiltration relative to vegetation water use). Soil characteristics for the study plots are presented in Table 2 under Results but some general initial indication of soil types, their spatial extent in the catchment and their relative susceptibility to erosion (e.g. expressed as Wischmeier K-values?) would not go amiss here (instead of the scattered reference made to soil characteristics across different sections). More importantly, give information on the age of the tree plantations (plots UT1, UT2, MT1 and MT2) and the height of the trees (important for assessing the erosive power of the raindrops, see previous comments on Discussion section) as well as some indication of tree densities in the various agroforestry plots (MT2-4), the width of the Casuarina strips in UT3, etc., etc. Likewise, photos of the respective plots could be added as Supplementary Material to give the reader a better impression, also in terms of plot sizes relative to terrace dimensions (were terrace beds back-sloping? If so, were adjacent upslope risers in plots MT1-4 included? (cf. Purwanto and Bruijnzeel, 1998); what was the nature of the terrace risers (grassed, weeds, stones?). NB: Table 1 still contains a number of plant names in Bahasa Indonesia (e.g. mahoni instead of mahogany) plus a number of typos (Albizia, manggo, dapap). *14, section 2.2 Rainfall: 'ombrometer' is an obsolete (colonial) term. Give dimensions of the rain gauge funnels and indicate whether these improvised gauges were calibrated against standard gauges to assess degree of under- or over-estimation of rainfall (rain-splash out of funnels during times of high intensity or effect of a broad rim on drop partitioning, etc.). Make clear in what plots the measurements represented rainfall (e.g. UT4?) or throughfall? (NB: adjust subsequent language in main text accordingly whenever discussing 'rainfall' if needed, e.g. in section 3.1, etc.). Add photos of position of gauges in Supplementary Materials since using only five rain/throughfall gauges per plot would seem inadequate given the

large variation in TF that is expected for such spatially variable vegetation? Also: two months duration, not three (lines 97-98) based on Figure 3. *15, line 104: awkward description of the runoff measurement system. Suggest to use the term 'divider system' instead and give maximum collection capacity for the two drums plus divider system in litres of water. NB: the volume given in Figure 2 for the drums (30 cm3) must be erroneous. Also, was the metal plate guiding the runoff to the drums sheltered against direct rainfall inputs? If not, runoff amounts will have been over-estimated somewhat. *16, line 109: strictly speaking, volume has the dimension of litres or cm3, not mm of water. You could simply give the volume in litres and divide by plot area in m2 and remain all right dimensionally. Suggest to remove the hyphens in d-I etc. in Equations 1 and 3 as they can be read as minus signs rather than hyphens. NB: second Dd-I in Equation 3 should read Dd-II. *17, lines 111-112: what was the efficiency of filtering your runoff samples using a newspaper compared to more conventional filters (e.g. Whatman or Millipore 0.45 $\mu$m)? Rijsdijk and Bruijnzeel (1990a) used simple coffee filters that were calibrated against conventional filtration. You may consider using a similar approach in future work. What about the sand fraction ending up in the first drum? Was the runoff water thoroughly stirred prior to taking the water sample? If so, inform the reader as such. *18, lines 122-125: did you take one block sample for bulk density measurement as suggested by the text or three? After all, you tested for differences in Table 2. How was particle density measured (by pycnometer?). *19, section 2.5.1. Canopy cover: it only becomes apparent in line 134 that the vegetation plots measured 20 m x 20 m; suggest to indicate this at the start of the plot descriptions. Lines 133-135: did you cover entire 5 m x 5 m areas with plastic/paper? References to Arumsari and Astutik are missing from reference list so cannot be checked (but might be in Bahasa Indonesia anyway and hence less accessible for most readers?). Line 136: suggest to replace CV in Equation 5 by another symbol to avoid confusion with coefficient of variation. NB1: one could also derive the canopy cover fraction from measurements of throughfall for small storms that do not saturate the canopy. The slope of the regression line between incident rainfall and free throughfall equals the gap fraction

(p), hence canopy cover fraction c equals 1 – p (Jackson, 1975). NB2: one wonders why direct observation of the contact cover fraction was not preferred to the more cumbersome (weighing/drying) litter mass approach? Contact cover fraction has been shown to be closely related to surface erosion rates in numerous cases (see e.g. Yu, 2005). *20, section 2.5.2. Understorey and litter: reference to Hairiah et al. is missing from the reference list (presumably the CIFOR publication). Indicate the number of replications used please. Using 50 cm x 50 cm would seem inadequate for understory measurements in the case of Lantana or Chromolaena shrubs. Were these present in the forest plots like they are in many plantations across Java? *21, section 2.5.3. Surface roughness: awkward formulation ('elevation', 'vertically'). Suggest rephrasing. *22, line 187: daily rainfall totals do not represent rainfall 'intensity' although you might refer to 'event intensity' if event durations are known. Lines 188-190: this belongs to Discussion rather than Results and is rather speculative anyway given the non-linearity of the rainfall-erosion relationship. Add discussion on what might constitute 'tolerable soil loss' in the study area given the rate of chemical denudation of andesitic ashes (= approximate rate of soil formation; Verheijen et al., 2009). See e.g. Bruijnzeel (1983) who determined the rate of chemical weathering for a high rainfall area with Inceptisols in Central Java at ca. 85 t km-2 yr-1. Given the difference in rainfall between his site and the Rejoso catchment a value of ca. 40 t km-2 yr-1 might be defendable, suggesting the tolerable soil loss might be as low as 0.4 t ha-1 yr-1? *23, lines 207-218 and 219-229: see main comments above. *24, line 231: what was the original slope steepness in the mid-stream area before bench terracing? Line 232: awkward formulation. Line 233: remove reference to seawater intrusion since not pertinent to present case? Line 234: 'erodible', not 'erosive'. Line 240: Liu, not Zhipeng. *25, line 245: in the middle and upper Rejoso watershed; line 247: keep erosion at acceptable levels? Line 248: gentle slopes associated with bench terracing or inherently gentle? If so, one wonders about the need for bench terracing. *26, lines 251-253: remove 'was' (four times); Remove Didik Suprayogo in lines 252-253. *27, references: standardize journal abbreviations, use of capitals, etc. Remove references not mentioned in text

(Alvarenga, Anache, Boongaling, Choto, Kellner, Teklay); add missing references given in main text including Astutik et al. 2015; Hairiah et al. 2005, Hoechstetter et al. 2008, Suprayogo et al. 2017, etc. *28, diagrams: Figure 1, add latitude/longitude indications; Figure 2, move to Supporting Materials as it does not add much or replace by a Photo?; correct the volume of the collector drums; Figure 3, use less awkward date indication; Figures 4, 5: indicate which panels refer to what land cover type; Figures 6-7: complete captions. Tables: add standard errors or coefficients of variation where appropriate.

Cited literature

Badu M., Nuberg I., Ghimire C.P., Bajracharya R.M., Meyer R.M. Negative trade-offs between community forest use and hydrological benefits in the forested catchments of Nepal's Mid-Hills. Mnt. Res. Dev. 39 (3): R22 – R32, 2019.

Bentley L., Coomes D.A. Partial river flow recovery with forest age is rare in the decades following establishment. Global Change Biol. 2020;00:1-16, 2019.

Bons C.A. Accelerated erosion due to clearcutting of plantation forest and subsequent Taungya cultivation in upland West Java, Indonesia. Int. Assoc. Hydrol. Sci. Publ. 192: 279 – 288, 1990.

Brown A.E., Western A.W., McMahon T.A., Zhang L. Impact of forest cover change on annual streamflow and flow duration curves. J. Hydrol. 483: 39 – 50, 2013.

Bruijnzeel L.A. The chemical mass balance of a small basin in a monsoonal environment and the effect of fast-growing plantation forest. Int. Assoc. Hydrol. Sci. Publ. 141: 229 – 239, 1983.

Bruijnzeel L.A. Hydrological functions of tropical forests: not seeing the soil for the trees? Agric. Ecosyst. Environ. 104: 185 – 228, 2004.

Cattan P., Bussiere F., Nouvellon A. Evidence of large rainfall partitioning patterns by banana and impact on surface runoff generation. Hydrol. Proc. 21: 2196 – 2205, 2007.

Coster C. Surficial runoff and erosion in Java. Tectona 31 (9/10): 613 – 728, 1938 (in Dutch with extended summary in English).

Filoso S., Bezerra M.O., Weiss K.C.B., Palmer M.A. Impacts of forest restoration on water yield: a systematic review. PLoS ONE 12 (8): e183210.

Friesen P., Park A., Sarmiento-Serrud A.A. Comparing rainfall interception in plantation trials of six tropical hardwood trees and wild sugar cane Saccharum spontaneum L. Ecohydrology 6: 765 – 774, 2013.

Ghimire C.P., Bruijnzeel L.A., Lubczynski M.W., Ravelona M., Zwartendijk B.W., van Meerveld H.J. Measurement and modeling of rainfall interception by two differently aged secondary forests in upland eastern Madagascar. J. Hydrol. 545: 212 – 225, 2017.

González-Martínez T.M., Williams-Linera G., Holwerda F. Understory and small trees contribute importantly to stemflow of a lower montane cloud forest. Hydrol Proc. 31: 1174 – 1183, 2017.

Hall R.L., Calder I.R. Drop size modification by forest canopies: measurements using a disdrometer. J. Geophys. Res. 98, D10: 18,465 – 18,470, 1993.

Holwerda F., Scatena F.N., Bruijnzeel L.A. Throughfall in a Puerto Rican lower montane rain forest: a comparison of sampling strategies. J. Hydrol. 327: 592 – 602, 2006.

Hutjes R.W.A., Wierda A., Veen A.W.L. Rainfall interception in the Tai Forest, Ivory Coast: application of two simulation models to a tropical system. J. Hydrol. 114: 259 – 275, 1990.

Jackson I.J. Relationships between rainfall parameters and interception by tropical forest. J. Hydrol. 24: 215 – 238, 1975. Lane P.N.J., Best A.E., Hickel K., Zhang L. The response of flow duration curves to afforestation. J. Hydrol. 310: 253 – 265, 2005.

Moreno-de las Heras M., Nicolau J., Merino-Martin L., Wilcox B.P. Plot-scale effects

on runoff and erosion along a slope degradation gradient. Water Resour. Res. 46, W04503.

Nespoulos J., Merino-Martin L, Monnier Y., Bouchet D.C., Ramel M., Dombey R., Viennois G., Mao Z., Zhang J.L., Cao K.F., Le Bissonnais Y., Sidle R.C., Stokes A. Tropical forest structure and understorey determine subsurface flow through biopores formed by plant roots. Catena 181: 104061.

Peña-Arancibia J.L., Bruijnzeel L.A., Mulligan M., van Dijk A.I.J.M. Forests as 'sponges' and 'pumps': assessing the impact of deforestation on dry-season flows across the tropics. J. Hydrol. 574: 946 – 963, 2019.

Prijono S., Midiyaningrum R., Nafriesa S. Infiltration and evaporation rate in different land use in the Bango watershed, Malang District, Indonesia. Int. J. Agric. Innov. Res. 3 (4): 1061 – 1067, 2014.

Purwanto E., Bruijnzeel L.A. Soil conservation on rainfed bench terraces in upland West Java, Indonesia: towards a new paradigm. Adv. GeoEcol. 31: 1267 – 1274, 1998.

Rijsdijk A. Evaluating sediment sources and delivery in a tropical volcanic watershed. Int. Assoc. Hydrol. Sci. 291: 16 – 23, 2005.

Rijsdijk A., Bruijnzeel L.A. Erosion, sediment yield and land-use patterns in the upper Konto Watershed, East Java, Indonesia. Part I. Introductory chapters. Konto River Project Comm. no. 18, Vol. 1. Konto River Project (ATA 206), Malang. DHV Consultants, Amersfoort, the Netherlands, 58 pp., 1990a.

Rijsdijk A., Bruijnzeel L.A. Erosion, sediment yield and land-use patterns in the upper Konto Watershed, East Java, Indonesia. Part II. Results of the 1987-89 measuring campaigns. Konto River Project Comm. no. 18, Vol. 2. Konto River Project (ATA 206), Malang. DHV Consultants, Amersfoort, the Netherlands, 150 pp., 1990b.

Rijsdijk A., Bruijnzeel L.A., Kukuh Sutoto, C. Production of runoff and sediment by

rural roads, trails and settlements in the Upper Konto catchment, East Java, Indonesia. Geomorphol. 87: 28 – 37, 2007.

Siles P., Vaast P., Dreyer E., Harmand J.M. Rainfall partitioning into throughfall, stemflow and interception loss in a coffee (Coffea Arabica L.) monoculture compared to an agroforestry system with Inga densiflora. J. Hydrol. 395: 39 – 48, 2010.

Stomph T.J., de Ridder N., Steenhuis T.S., van de Giesen N.C. Scale effects of Hortonian overland flow and rainfall-runoff dynamics: laboratory validation of a process-based model. Earth Surface Proc. Landf. 27: 847 – 855, 2002.

Taniguchi M., Tsujimura M., Tanaka T. Significance of stemflow in groundwater recharge. 1: Evaluation of the stemflow contribution to recharge using a mass balance approach. Hydrol. Proc. 10: 71 – 80, 1996.

Toohey R.C., Boll J., Brooks E.S., Jones J.R. Effects of land use on soil properties and hydrological processes at the point, plot, and catchment scale on volcanic soils near Turrialba, Costa Rica. Geoderma 315: 138 – 148, 2018.

Van Dijk A.I.J.M., Bruijnzeel L.A. Runoff and soil loss from bench terraces. 1. An event-based model of rainfall infiltration and surface runoff. Europ. J. Soil Sci. 55: 299 – 316, 2004a.

Van Dijk A.I.J.M., Bruijnzeel L.A. Runoff and soil loss from bench terraces. 2. An event-based erosion process model. Europ. J. Soil Sci. 55: 317 – 334, 2004b.

Van Noordwijk M., Tanika L., Lusiana B. Flood risk reduction and flow buffering as ecosystem services – Part 1: Theory on flow persistence, flashiness and base flow. Hydrol. Earth Syst. Sci. 21: 2321 – 2340, 2017a.

Van Noordwijk M., Tanika L., Lusiana B. Flood risk reduction and flow buffering as ecosystem services – Part 2: Land use and rainfall intensity effects in Southeast Asia. Hydrol. Earth Syst. Sci. 21: 2341 – 2360, 2017b.

[Figure]

Verheijen F.G.A., Jones R.J.A., Rickson R.J., Smith C.J. Tolerable versus actual soil erosion rates. Earth Sci. Rev. 94: 23 – 38, 2009.

Wiersum K.F. Effects of various vegetation layers in an Acacia auriculiformis forest plantations on surface erosion in Java, Indonesia. In: S.A. El-Swaify, W.C. Moldenhauer, A. Lo (Eds.), Soil Erosion and Conservation. Soil Conservation Society of America, Ankeny, IA, U.S.A., pp. 79 – 89, 1985.

Yu, B.F. Process-based erosion modelling: promises and progress. In: M. Bonell, L.A. Bruijnzeel (Eds.), Forests, Water and People in the Humid Tropics. Cambridge Univ. Press, Cambridge, U.K., pp. 790 – 810, 2005.

Zhang J., Bruijnzeel L.A., van Meerveld H.J., Ghimire C.P., Tripoli R., Pasa A., Herbohn J. Typhoon-induced changes in rainfall interception losses from a tropical multi-species 'reforest'. J. Hydrol. 568: 658 – 675, 2019.

Zhang M.F., Liu N., Harper R., Li Q., Liu K., Wei X., Ning D., Hou Y., Liu S. A global review on hydrological response to forest change across multiple spatial scales: importance of scale, climate, forest type and hydrological regime. J. Hydrol. 546: 44 – 59, 2017.

Zion Klos P., Chain-Guadarrama A., Link T.E., Finegan B., Vierling L.A., Chazdon R. Throughfall heterogeneity in tropical forested l

---

## Referee Comment (RC2) · Anonymous Referee #2 · 8 Apr 2020

The study by Suprayogo et al. addresses a relevant topic and will be of interest to the hydrologic community. However, the work lacks the quality that is expected from a HESS publication. Some of the major issues are - ) Introduction section is quite weak and doesn't offer enough lit review to gauge the novelty of the work; 2) research questions are vague and need to be carefully thought through; 3) Several method related details are unclear, and the statistical aspect of the work requires more effort. 4) Discussion lacks critical thinking, and authors need to carefully think about the key messages that they want to discuss and convey. Lastly, the work needs clarity and organization throughout the sections. I would recommend the authors to revise the manuscript substantially and resubmit.

[Figure]

Section specific comments are below, and the line specific comments are in the attached pdf.

Abstract – Too much unnecessary detail and the relevant information is missing. Most of the abstract focuses on introduction and methods and relatively little text on interpretations and implications of the work.

Introduction – The authors need to highlight the novelty and research gap in their work. What is the research gap that you want to address? Is it about the volcanic landscape, erosion in a forest to open-field-agriculture continuum or, both? How many studies have done similar work in a volcanic landscape? If any, what were their findings? Research questions could use some clarity and can better. In question 1, "...limit infiltration below the required rate" what do you mean by required rate? How do you define it? Please be more quantitative about it. Similarly, "infiltration friendly" is also a vague term.

Generally, the topical sentence of a paragraph introduces an idea and rest of the texts expand on it. In the manuscript, several ideas have been introduced in the same paragraph, resulting in an incoherent passage with no clear message. This is true for many paragraphs throughout the draft. Lastly, the entire section could use better organization.

Methods Several method related details are unclear, and assumptions made during measurements have not been clearly laid out. How many places did you measure rainfall/through fall underneath the canopy? How do we account for the spatial variability in rainfall due to heterogeneity in canopy cover? The method used for estimating canopy cover is unclear, and the citation is missing in the References section.

"A probability level of 0.05 was set for rejecting null-hypothesis of no difference in tests of statistical.." what were the hypotheses that you tested? No hypothesis has been shared in the manuscript. Are you referring to LSD tests? In the field conditions, most of the key drivers will "interact" in response to rainfall. The regression model with a

single explanatory variable may not be an ideal approach here. Would it be possible to use models that incorporate some interaction effect between the explanatory variables? Also, did you explore the correlation among explanatory variables? If you are really interested in exploring the influence of an individual exploratory variable on erosion, please use partial correlations that allows to control the effect of other related variables on responses. Results and Discussion The discussion section could use some critical thinking, especially about the key messages that you want to convey. Most of the discussion revolves around the major drivers of responses. The section barely discusses questions 1 and 3. Lastly, you want to show how the work advances our current understanding of runoff and erosion processes in the volcanic region.

Line specific comments are annotated in the attached pdf.

Please also note the supplement to this comment:
https://www.hydrol-earth-syst-sci-discuss.net/hess-2020-2/hess-2020-2-RC2-supplement.pdf

**Supplement:**

[revised manuscript text omitted]

---

## Author Comment (AC1) · 17 May 2020

**1. General comment**

Comment 1 The paper by Suprayogo et al. describes the results of a short-term (ca. two months) field experiment aiming to identify which of the dominant land uses in the middle and upper parts of the 633 km2 upland Rejoso catchment in the volcanic uplands of East Java may be considered 'infiltration-friendly' under the prevailing rainfall intensities.

Feedback comment 1 Yes, we do agree. In the Rejoso watershed, degradation has

become an issue and land use on the slopes is suspected to be one of the causes. Practical answers are needed

Comment 2 In addition, the study seeks to identify which readily measured vegetation and surface characteristics may be used to define the critical threshold values associated with such 'infiltration-friendly' land uses.

Feedback comment 2 That is right, we stated in line 43-64

Comment 3 The rationale for this work partly us in an observed decline in the quantity and quality of water resources in the study area believed to reflect changes in land use and land cover in the Rejoso basin as well as increased extraction of groundwater for irrigated rice cultivation in the lower parts of the catchment.

Feedback comment 3 Yes, some video's explaining the situation are available was made by Danon

Comment 4 The authors rightly point out that much of the debate on land cover and catchment water yield and/or stream-flow regimes focuses on forested versus agricultural uses of the land, whereas preciously little is known on the comparative ability of such 'intermediate' land-use types as agroforestry systems in terms of maintaining soil infiltration capacity, groundwater recharge and dry-season flows under seasonal tropical conditions (Toohey et al., 2018; Nespoulos et al., 2019).

Likewise, most global reviews of the literature on 'forests and water' have concentrated on the changes in total water yield associated with forest removal or addition, interpreting the observed changes in flow primarily in terms of increases or decreases in vegetation water use (evapotranspiration, ET) as a function of forest type or climate, but leaving the effects on flow regime by potential changes in surface conditions essentially non-analyzed (e.g. Zhang et al., 2017; Filoso et al., 2017; Bentley & Coomes, 2019). Only a few studies have paid explicit attention to changes in seasonal streamflow regime after removing or adding trees (e.g. Lane et al., 2005; Brown

et al., 2013; cf. Van Noordwijk et al., 2017ab). In view of the fact that such changes in streamïnĆow regime will reïnĆect both changes in ET and in inïnĄltration after the change in land use under given climatic conditions (Bruijnzeel, 2004; Peña-Arancíbia et al., 2019) the paper by Suprayogo et al. is to be welcomed in principle in that it documents inïnĄltration (at the runoff plot scale) and erosion rates for a series of agroforestry systems (both terraced and non-terraced), rain-fed crops (mostly maize) and forest plantations (pine and mahogany) that may be considered typical for Java's densely populated uplands.

Feedback comment 4 In the Rejoso watershed, the main issue that occurred was land cover changes from forest to agricultural uses and had a very significant impact on catchment water yield and / or stream flow. The function of agroforestry as a trade-off between production needs and watershed buffer hydrology has not been much studied in the tropics. This research study is unique to contribute knowledge of how agro-forestry systems in terms of maintaining soil infiltration capacity, groundwater recharge and dry-season ïnĆows under tropical conditions. For that, thank you for your comment on this research novelty.

Comment 5 Moreover, the paper marks a valiant attempt at identifying some of the key vegetation and surface characteristics controlling inïnĄltration rates, runoff production (i.e. overland ïnĆow) and surface erosion rates using low-cost / low-tech approaches for measuring rainfall, runoff and site characteristics.

Feedback comment 5 This is true, that we designed research equipment using low-cost equipment and simple equipment, but we design it by following the principles of a measurement approach that can be scientifically justified. This is done with the consideration that the measurement is done in a remote location. With the low cost equipment, we do hope that will not attract the attention of the surrounding commu-nity about equipment that has low economic value, so it is safe not to get damage or vandalism from one or two irresponsible community members.
Comment 6 The paper's key fidings include strongly negative relationships between canopy cover fraction and either plot-scale runoff coefficient (Fig. 6) or surface erosion (Fig. 7), and to a lesser extent between the latter and surface litter amounts while understory biomass seems less important. However, individual land-use types deviated from this general pattern. For example, the 'young' and 'old' production forest plots (UT1 and UT2) had the same runoff coefficients (13-14%) but the young forest exhibited a far smaller soil loss than the older forest, despite having a more open canopy (by 12%, difference not signïficant), a much lower litter mass (2.0 vs. 9.2 t ha-1) and a much steeper slope (50-60% vs. 30-40%) (Tables 3 and 1). Likewise, agroforestry plots MT2 and MT3 exhibited the same (high) runoff coefficient (37-38%) but differed by a factor of 2.3 in soil loss with the amount of litter on the ground being the same (Tables 3 and 1). Such findings suggest that soil characteristics (as opposed to surface or vegetation variables) likely play a role as well and perhaps deserve to be included in any predictive equations (e.g. SOC?; applicable in the case of MT2 vs. MT3 but not for UT1 vs. UT2 (Table 2)) (Table 2)). At any rate, it would be good to include such considerations explicitly in the Discussion section.

Feedback comment 6 Thank you for the novelty's emphasis on this research. We use these comments to revise the descriptions in the results and discussion.

Comment 7 On the downside, the paper gives the distinct impression of having been put together in some haste and the often highly concise text leaves much to be guessed (or derived) by the reader. This holds especially true for the sections describing the study area and methods, but also for the Results and Discussion. For example, the study area description effectively consists of a Table describing the landcover types for the eight examined locations (Table 1) but fails to give basic location or climatic information, or a proper description of vegetation characteristics (e.g. tree height – important to assess the erosive power of crown drip) or the nature of the terraces of the mid-stream research plots (notably whether the plots included terrace risers or terrace beds only).

Feedback comment 7 The basic location we will give description as suggested in specific comment no 12. The basic climate information, we have daily data for up-stream and mid-stream for long term period (12 year). We will presenting the average data for this period at monthly basic within a year period. Regarding the description of vegetation characteristics, we measured the tree height and we will presenting into this manuscript. We presenting the tree height in Table 1 in supplement note no 1.

The nature of terraces is considered as bench terrace sloping outward. The measurement of runoff and soil erosion in the terrace bed and do not considered terrace riser.

Comment 8 As to the methods used, it is not clear – inter alia – in which plots 'rainfall' truly represents incident rainfall or rather crown drip (throughfall) (line 101); what size or type of funnel was used for the improvised rain gauges and what the catch efficiency of these gauges was compared to that of a standard rain gauge (cf. Ghimire et al., 2017; Zhang et al., 2019); how the collection drums with a stated capacity of 30 cm3 (!) (Fig. 2) could accommodate the runoff produced by 12 m2 plots with runoff coefficients of up to 41-64% (Table 4); how coarse sediment eroded from the plots was accounted for or what the effect of filtering runoff samples through 'newsprint' (lines 111-112) was on the magnitude of sediment concentrations obtained compared to more conventional filtration methods; how particle density was determined (line 124); or how many replications were used to determine undergrowth or litter mass (lines 140-141), etc.

Feedback comment 8 We measured throughfall by using simple rain-gauge with mineral bottle combined with 30 cm diameter of plastic funnel. This diameter funnel is considered standard of rain gouge. Therefore, We do not measured catch efficiency. Each plot, we installed 5 rain-gauge randomly.

Two collection drums with a capacity of 30 dm3 (was not 30 cm3) was used to collect the runoff and sediment, where the first drum was channeling into 13 channel PVC pipes with equally hole size and the levelling position and one of others connected with

second drum. The volume of water come out from each hole we calibrate to get the water volume proportion enter into second drum. The potential capacity of the runoff collector then can be = (25 dm3 * 13) + 25 dm3) = 350 dm3. If the plot size by 12 m2 and assumes all the rainfall become runoff then this runoff collector can collect = 350 dm3/1200 dm2 = 0.292 dm = 292 mm rain event.

The course sediment was trapped in first drum. During the sediment sampling, the sediment and water in each drum was stirred homogeneously and then take one liter of sediment samples. We filtering the sediment sample using 'newsprint'. The water through a filter 'newsprint' was relatively clear. Therefore, we consider that 'newsprint' filter is considered effective to trap the sediment. We do not calibrate between 'newsprint' filter with standard filter. The particle density was measured by using pycnometer method.

We use this scheme below (presenting in supplement note 2 to measure the undergrowth or litter mass. This scheme based on based the protocol by Hairiah K, Ekadinata A, Sari RR, Rahayu S. Measurement of Carbon Stock: from land level to landscape. Practical instructions. Second edition. Bogor, World Agroforestry Centre, ICRAF SEA Regional Office, University of Brawijaya (UB), Malang, Indonesia 110 p.[Indonesia] http://apps.worldagroforestry.org/sea/Publications/files/manual/MN0049-11.pdf. 2011

Comment 9 Ibidem for the Results section: e.g. the main results for soil properties are described in 1.5 lines only (lines 162-163);

Feedback comment 9 The two soil groups chosen to represent "Infiltration-Friendly" Land Uses, namely Inceptisol and Andisol do have different characteristics. We have soil texture every 10 cm soil depth at a depth of 00-50 cm. The bulk density, particle density, porosity macroporoity and soil orgnic mater observations every 10 cm soil depth at a depth of 00-30 cm from the surfaces, showed that the two soil groups have deep soil profiles. The striking difference between the two soil groups is finer soil texture, higher bulk density, lower macro-porosity and lower soil infiltration in the Inceptisol than in the Andisol. The data we presented in supplement Note 3, and we will added those data and give more comprehensive description and discussion related with the soil characteristic to answer the first research question.

Comment 10 key Tables 2-4 give averages or period totals only but no standard errors or coefficients of variation as a measure of within-plot variability despite the fact that the large variation in rainfall (throughfall?) totals between adjacent (?) plots (e.g. 300 mm for plots MT4 and MT2 in Table 4) suggested major spatial variability;

Feedback comment 10 Regarding Tables 2-4 we used Fisher's LSD test (p<0.05). to represent the standard errors or coefficients of variation.

Comment 11 The captions to key diagrams 4 and 5 which count 8 panels each do not explicitly specify which panel refers to which plot in any descriptive way other than their relative position in the list of plots in Table 1.

Feedback comment 11 We revised as follow: Figure 4. The relationship between surface runoff / rainfall ratio and the amount of rainfall in production forest and agroforestry systems in (a) Upstream Rejoso Watershed, under (a.1) 55% canopy cover of Pine based of old production forest, (a.2) 40% canopy cover of Pine based of young production forest, (a.3) 5% canopy cover of Cemara based of Agroforestry with Cabbage crop, (a.4) 0% tree canopy cover of Arable land (maize crop) ; (b) Midstream Rejoso Watershed under (b.1) 87 % canopy cover of Pine/ mahogany based of old production forest, (b.2) 75% canopy cover of Coffee-based agroforestry, (b.3) 52 % canopy cover of Clove based agroforestry, (b.4) 26% canopy cover of mix tees-crop based agroforestry.

Figure 5: Soil erosion in relation to daily rainfall rates in production forest and agroforestry (a) Upstream Rejoso Watershed, under (a.1) 55% canopy cover of Pine based of old production forest, (a.2) 40% canopy cover of Pine based of young production forest, (a.3) 5% canopy cover of Cemara based of Agroforestry with Cabbage crop, (a.4) 0% tree canopy cover of Arable land (maize crop) ; (b) Midstream Rejoso Watershed under (b.1) 87 % canopy cover of Pine/ mahogany based of old production forest, (b.2) 75% canopy cover of Coffee-based agroforestry, (b.3) 52 % canopy cover of Clove based agroforestry, (b.4) 26% canopy cover of mix tees-crop based agroforestry.

Comment 12 The Discussion section is rather basic and does not address such critical issues as the improvised nature of the rain gauges and the neglecting of stemflow as a possible input of water to the soil (which would lead to under-estimation of 'rainfall' and hence over-estimation of runoff coefiňĄcients), as well as the scale and ability of the plot-based measurements to represent the entire hilllslope's hydrological functioning (see below for details and related issues)).

Feedback comment 12 Thank you for the emphasis on this issues. We will use these comments to revise the descriptions in the results and discussion.

Comment 13 Furthermore, the reference list – although remarkably up to date with more than 60% of cited references published between 2017 and 2019 – misses at least six references that are cited in the text and contains another six that are listed all right but do not show up in the text (see below under specific comments).

Feedback comment 13 We have been revised as list the in supplement note 4

Comment 14 On a related note, rather than to refer largely to recent studies in such very different environments as the semi-arid loess plateau in China (Zhao et al., 2019; Liu et al., 2018 – erroneously referred to as Zhipeng et al., 2018) the paper would arguably have benefitted from the inclusion of related studies in the volcanic uplands of West and East Java for added perspective. Examples include the effect of trampling/footpaths on runoff production and erosion from upland fields (Bons, 1990; Rijsdijk et al., 2007; cf. Badu et al., 2019); the role of terrace risers versus terrace beds vis á vis runoff and sediment production (Purwanto and Bruijnzeel, 1998; Van Dijk & Bruijnzeel, 2004ab); the potential role of stemflow in the generation of localized infiltration or overland flow (high stemflow fractions reported for bamboo, bananas, shaded coffee, tall grasses and understory shrubs (Taniguchi et al., 1996;

Cattan et al., 2007; Siles et al., 2010; Friesen et al., 2013; González-Martínez et al. 2016)); as well as a more holistic discussion of the interactions between canopy cover fraction, understory, leaf litter and hydrological processes (Wiersum, 1985; cf. Prijono et al., 2014; even Coster, 1938) – not only in terms of amounts of water involved but also the erosive power of the rain / crown drip as a function of falling height and leaf type (Wiersum, 1985; Hall and Calder, 1993).

Feedback comment 14 Thank you for the enrichment of references in the discussion that further clarifies the conditions of the tropics. We will include these references in the discussion.

Comment 15 In contrast to what is stated in the paper (lines 209-211), rainfall interception does not reduce the erosive power of the rain. Rather, a tree canopy enhances it because of the typically larger drop size of crown drip compared to incident rainfall for terminal fall velocities (Wiersum, 1985), with broad-leaved species producing larger drops than do conifers (Hall and Calder, 1993). Likewise, the few measurements of throughfall intensities under humid tropical conditions suggest these to be very similar to those of incident rainfall (e.g. Hutjes et al., 1990). As such, the presumed effect on delaying the onset of overland flow (line 211) would seem rather negligible. Similarly, based on the lack of correlations with surface runoff or erosion (Figs. 6 and 7) the role of understory vegetation is considered to be minor.

Feedback comment 15 Thank you for expanding knowledge about the process of the role of canopy in controlling surface runoff and erosion. We will include this consideration in the discussion.

Comment 16 Yet Nespoulos et al. (2019) emphasized the importance of a well-developed understory for the development of macroporosity and preferential pathways in tropical plantations. In addition, the discussion could use a paragraph on the importance of including both infiltration and total ET (not just interception loss) for a proper assessment of the effect of land cover on groundwater recharge.

Feedback comment 16 We have measurements of macroporosity (based on methylene blue infiltration patterns) data and we will add this to the manuscript. We will refer to this in the revised discussion.

Comment 17 Other points not considered in the Discussion include such aspects like (a) the need for an adequate number of throughfall gauges to quantify net precipitation beneath such spatially highly variable vegetation types as some of the studied agroforestry systems (cf. Holwerda et al., 2006; Zion Klos et al., 2014); (b) the role of auto-correlation in the presented relations between runoff coefiňǍcient (i.e. runoff/ rainfall) and rainfall, (c) the possible role of (high) short-term rainfall intensities as opposed to the presently used daily totals in determining measured amounts of surface runoff and eroded soil (cf. Wischmeier's EI30 index), (d) the effect of the small size of the runoff plots used (2 m x 6 m) on measured amounts of surface runoff vis á vis their representativeness for hydrological functioning at the hillslope scale (variations in slope steepness, reinfiltration on less steep foot-slopes in the case of the upper plots; cf. Stomph et al., 2002; Moreno-de las Heras et al., 2010), and (d) what might constituted a plausible value of 'tolerable soil loss' for the study area (Verheijen et al., 2009; cf. Bruijnzeel, 1983; see also specific comment *22 below for a possibly locally valid estimate).

Feedback comment 17 Thank you for the enrichment of references in the discussion (a) the importance of the amount of throughfall gauges to quantify net precipitation, (b) the role of auto-correlation in the presented relations between runoff coefficient (ie runoff / rainfall) and rainfall (c) the possible role of ( high) short-term rainfall intensity as opposed to the presently used daily totals in determining measured amounts of surface runoff and eroded soil; (d) the effect of the small size of the runoff plots used for understanding the hydrological functioning at landscape scales, (e) considering "tolerable soil loss" in the discussion. We will include these considerations and references in the discussion.

Comment 18 On the basis of the above considerations my recommendation for this

manuscript can only be rejection in its present form but allow resubmission if the main points raised in the previous paragraphs can be addressed satisfactorily. After all, the paper addresses an important topic and new information on infiAltration behaviour of agroforestry systems is to be welcomed.

Feedback comment 18 Thank you for the enrichment of references in the discussion that further clarifies the conditions of the tropics. We will include these references in the discussion.

2. Specific comments

Comment 1 *1, Title: you may want to use inverted commas for 'infiltration friendly' in the title as well.

Feedback comment 1 Using inverted commas's in the title would lead to:'Infiltration-Friendly' Land Uses for Climate Resilience on Volcanic Slopes in the Rejoso Watershed, East Java, Indonesia We considered this suggestion, but think that it will create more confusion that it solves. If the reader gets interested in what this term means, we invite them to read the abstract and paper in which it is explained.

Comment 2 *2, line 20: based on the time line in Figure 3 (6 March–1 May 2017) a period of nearly two months would be more accurate than 'three months'

Feedback comment 2 We quantified infiltration and erosion in three replications per land use category over a period of nearly two months (one-fourth of mean annual rainfall), with 6-13% of rainfall with intensities (51-100 mm day-1).

Comment 3 *3, line 21: strictly speaking, when using daily rainfall and surface runoff totals to obtain net infiltration amounts one cannot refer to the latter as infiltration rates.

Feedback comment 3 We related soil infiltration to plot-level characteristics across the land use systems and found statistically significant relations with . . . . . .(Note we replace instead of "infiltration rate" to become "soil infiltration")

Comment 4 . *4, line 24: 'preceding water use' is unnecessarily vague; suggest replacing by 'soil moisture status' because moisture values are governed by the interplay of rainfall/drip, evaporation and soil water uptake, not just vegetation water uptake.

Feedback comment 4 . . ...with tree canopy cover (likely based on combined effects of interception, soil moisture status and effects on soil), understory cover, amount of litter, and soil surface roughness.

Comment 5 *5, line 24: relations with understory biomass or surface micro-topographic variation were not strong or absent (Figs. 6 and 7).

Feedback comment 5 We related soil infiltration to plot-level characteristics across the land use systems and found statistically significant relations with tree canopy cover (likely based on combined effects of interception, soil moisture status and effects on soil) and amount of litter. The soil infiltration has found statistically significant with surface micro-topographic variation under forest-agroforestry systems (midstream), but has not under vegetable cultivation in steep to very steep lands. There is no significant relation between soil erosion and understory biomass of land use system.

Comment 6 *6, line 28: an average runoff coefiňĄcient of 62% (actually 64% in Table 4) is exceedingly high and more representative of compacted dirt roads and yards in the area than actively worked agricultural iňĄelds (see e.g. Rijsdijk et al., 2007). Such high values might suggest rainfall inputs may well have been under-estimated. Worth checking!

Feedback comment 6 For a tree canopy cover in the range 20-80%, erosion rates were relatively low, but surface runoff increased to 36 to 64% of rainfall. Differences in soil type influenced the thresholds, as the areas' Inceptisols have lower intrinsic porosity than Andisols. However, we check again the data.

Comment 7 *7, line 29: with porosities of 55-62% and bulk densities of 0.9-1.1 g cm3 (Table 2) the soils of the mid-stream sites are not particularly dense. Rather, one would

think of crusting or slaking as a potential cause for low apparent inifiltration?

Feedback comment 7 The data of soil Bulk Density that we presented was a top soil data. Forest or agroforestry land use in mid-stream can provides input of organic material and then can have relatively high microorganism activity. This such condition can stimulate low soil bulk density values. Infiltration in the mid-stream is low because the subsoil layer has a fine texture and high bulk density. To prove this, we added data not only on top soil layers but also sub-soil and infiltration data measured with a double ring infiltrometer.

Comment 8 *8 lines 41-42: this sentence seems to fall out of the blue; since the cited reference concerns a global review of the literature on surface erosion it might be possible that the authors meant Wiersum (1985) instead which documents the role of understory and litter layer with regards to surface runoff and erosion in an Acacia auriculiformis plantation in West Java, not pines. Incidently, drip from a pine canopy is less erosive than that from broad-leaved species like Eucalyptus and, especially Teak (Hall and Calder, 1993).

Feedback comment 8 Thank you for your note. We will revised on this part

Comment 9 *9, line 52: the sentence on inifiltration recovery seems out of place here and had perhaps better be moved to the Discussion section.

Feedback comment 9 Thank you for your note. We will moved to discussion section.

Comment 10 *10, line 54: whilst the inifluence of a very extensive and aerodynamically trough forest cover on rainfall may have an effect on downwind rainfall amounts (as opposed to 'events'), it would seem inappropriate to mention this in the present context of on-site water dynamics. Suggest to leave out this aspect.

Feedback comment 10 Thank you for this comment. We will leaved out this aspect.

Comment 11 *11, lines 62-64: unclear why Climate Resilience is written with initial capitals?; also, relation between iňĆow persistence metrics and peakiňĆow transmission

(routing? percolation?) is not instantly clear.

Feedback comment 11 In watersheds that provide a perfect buffering river flow might theoretically be constant every day, but in practice a 'flow persistence' metric of about 0.85 is hard to surpass (van Noordwijk et al., 2017). Flow buffering is essential for Climate Resilience (Aduah et al., 2017; Shannon et al., 2019) and high flow persistence metrics are desirable, as they directly relate to peak flow transmission (van Noordwijk et al., 2017).

Comment 12 *12, lines 77-78: soil fertility and agricultural productivity may be maintained sufficiently on these deep volcanic deposits even if surface erosion rates are high. Also, previous research on sediment production in similar terrain nearby in East Java (Rijsdijk and Bruijnzeel, 1990ab; Rijsdijk, 2005) has shown that contributions from rain-fed agricultural fields made up a comparatively minor proportion of overall annual sediment yield at the operational catchment scale with roads, paths, settlements contributing significant amounts each.

Feedback comment 12 Thank you for your comment. The research of Rijsdijk and Bruijnzeel, 1990ab; and Rijsdijk, 2005, will be added in this part.

Comment 13 *13, section 2.1, Study area: suggest to give a proper basic description of site locations (place names, latitude, longitude, elevations) along with information on the main environmental conditions, notably (a) annual rainfall totals and the agroclimatic classification of the two main sites in terms of rainfall seasonality (e.g. Oldeman climate types D3 and C2 for the middle and upper zones of the catchment?), (b) prevailing rainfall intensities (e.g. based on the authors their own measurements or the iso-erodent map of Java ?, and perhaps (c) FAO reference evapotranspiration (to help assess the importance of differences in infiltration relative to vegetation water use). Soil characteristics for the study plots are presented in Table 2 under Results but some general initial indication of soil types, their spatial extent in the catchment and their relative susceptibility to erosion (e.g. expressed as Wischmeier K-values?) would not

go amiss here (instead of the scattered reference made to soil characteristics across different sections). More importantly, give information on the age of the tree plantations (plots UT1, UT2, MT1 and MT2) and the height of the trees (important for assessing the erosive power of the raindrops, see previous comments on Discussion section) as well as some indication of tree densities in the various agroforestry plots (MT2-4), the width of the Casuarina strips in UT3, etc., etc. Likewise, photos of the respective plots could be added as Supplementary Material to give the reader a better impression, also in terms of plot sizes relative to terrace dimensions (were terrace beds back-sloping? If so, were adjacent upslope risers in plots MT1-4 included? (cf. Purwanto and Bruijnzeel, 1998); what was the nature of the terrace risers (grassed, weeds, stones?). NB: Table 1 still contains a number of plant names in Bahasa Indonesia (e.g. mahoni instead of mahogany) plus a number of typos (Albizia, manggo, dapap).

Feedback comment 13 We will revised as your suggestion. Some example revision will be as in supplement note 5.

Comment 14 *14, section 2.2 Rainfall: 'ombrometer' is an obsolete (colonial) term. Give dimensions of the rain gauge funnels and indicate whether these improvised gauges were calibrated against standard gauges to assess degree of under- or over-estimation of rainfall (rain-splash out of funnels during times of high intensity or effect of a broad rim on drop partitioning, etc.). Make clear in what plots the measurements represented rainfall (e.g. UT4?) or throughfall? (NB: adjust subsequent language in main text accordingly whenever discussing 'rainfall' if needed, e.g. in section 3.1, etc.).

Add photos of position of gauges in Supplementary Materials since using only fi̧ve rain/throughfall gauges per plot would seem inadequate given the large variation in TF that is expected for such spatially variable vegetation? Also: two months duration, not three (lines 97-98) based on Figure 3.

Feedback comment 14 We revised as: "Rain gauges were installed in four observation locations (with adjacent erosion plots) upstream and four observation locations

midstream of the Rejoso watershed. Each plot, the rainfall was measured with 5 replications. The rain gauges consisted of 30 cm diameter of funnel and bottle with a volume of 1.5 dm3 placed 120 cm above the soil surface and below the tree canopy with bamboo as a support. Rainfall was observed every day during two months of the rainy season, from March to May 2017".

We will add photos of position of gauges in supplementary materials.

We revised as two months duration.

Comment 15 *15, line 104: awkward description of the runoff measurement system. Suggest to use the term 'divider system' instead and give maximum collection capacity for the two drums plus divider system in litres of water. NB: the volume given in Figure 2 for the drums (30 cm3) must be erroneous. Also, was the metal plate guiding the runoff to the drums sheltered against direct rainfall inputs? If not, runoff amounts will have been over-estimated somewhat. *16, line 109: strictly speaking, volume has the dimension of litres or cm3, not mm of water. You could simply give the volume in litres and divide by plot area in m2 and remain all right dimensionally. Suggest to remove the hyphens in d-I etc. in Equations 1 and 3 as they can be read as minus signs rather than hyphens. NB: second Dd-I in Equation 3 should read Dd-II.

Feedback comment 15 We revised as: "In each plot, the water flow out through 13 holes of PVC pipes on drum-I was calibrated to be equal volume of water in each hole before runoff measurement.

Revision of Figure 2 is presented in supplement note 6.

In each plot, the water flow was collected into two collection drums with a capacity of 30 dm3. where the first drum has divider system with channeling into 13 channel PVC pipes with equally hole size and the levelling position and one of others connected with second drum. The volume of water come out from each hole we calibrate to get the water volume proportion enter into second drum. The potential capacity of the runoff

collector then can be = (25 dm3 * 13) + 25 dm3) = 350 dm3. Runoff samples at each plot were collected on every day at which rain occurred during the measurement period by measuring the water depth in each drum. The amount of runoff in each rain even was calculated using eq. (1) and eq. (2):

$R\_t= V\_{(d-I)}+(13* V\_{(d-II)})$ [1]

$V\_d=1000* (D*L*W)$ [2]

Where Rt is total runoff (dm3); Vd is the water volume in drum I and II (dm3), L= Length and W width of drum (cm), D is the water depth in each drum (cm). The total runoff then converted as a mm unit by dividing areas of the plot (2 m x 6 m).

The soil erosion in each rain even was determined by collecting 1 dm3 of runoff-sediment in each drum. The sample was filtered with "newsprint" and dried in the oven with temperature 105oC to get the weight of sediment (S). The soil erosion in each rain even was calculated using eq. (3):

$E=((V\_{(d-1)}*S)+(13*(V\_{(d-2)}*S)))*(ãĂŰ10ãĂŮ^{(-2)}/A)$ [3]

Where E is soil erosion (ton ha-1); S is sediment (g dm-3), A is the areas of plot (m2).

The metal plate guiding the runoff to the drums sheltered against direct rainfall inputs.

Comment 16 *17, lines 111-112: what was the efficiency of filtering your runoff samples using a newspaper compared to more conventional filters (e.g. Whatman or Millipore 0.45 _m)? Rijsdijk and Bruijnzeel (1990a) used simple coffee filters that were calibrated against conventional filtration. You may consider using a similar approach in future work. What about the sand fraction ending up in the first drum? Was the runoff water thoroughly stirred prior to taking the water sample? If so, inform the reader as such.

Feedback comment 16 We filtered the sediment sample using 'newsprint'. The water through a filter 'newsprint' was relatively clear. Therefore, we consider that 'newsprint'

filter is considered effective to trap the sediment. We did not calibrate between 'newsprint' filter with standard filter (e.g. Whatman or Millipore 0.45 _m) as part of this research, but will look for additional data from earlier studies in our lab The course sediment (sand fraction) was trapped in first drum. During the sediment sampling, the sediment and water in each drum was stirred vigorously before taking one liter of sediment samples.

Comment 17 *18, lines 122-125: did you take one block sample for bulk density measurement as suggested by the text or three? After all, you tested for differences in Table 2. How was particle density measured (by pycnometer?).

Feedback comment 17 We take one block sample (20 cm x 20 cm x 10 cm = 4000 cm3) for bulk density measurement. Particle density measured by pycnometer method.

Comment 18 *19, section2.5.1. Canopy cover: it only becomes apparent in line 134 that the vegetation plots measured 20 m x 20 m; suggest to indicate this at the start of the plot descriptions.

Lines 133-135: did you cover entire 5 m x 5 m areas with plastic/paper? References to Arumsari and Astutik are missing from reference list so cannot be checked (but might be in Bahasa Indonesia anyway and hence less accessible for most readers?).

Line 136: suggest to replace CV in Equation 5 by another symbol to avoid confusion with coefficient of variation. NB1: one could also derive the canopy cover fraction from measurements of throughfall for small storms that do not saturate the canopy. The slope of the regression line between incident rainfall and free throughfall equals the gap fraction (p), hence canopy cover fraction c equals 1 – p (Jackson, 1975). NB2: one wonders why direct observation of the contact cover fraction was not preferred to the more cumbersome (weighing/drying) litter mass approach? Contact cover fraction has been shown to be closely related to surface erosion rates in numerous cases (see e.g. Yu, 2005).

Feedback comment 18 We will revised that the vegetation plots measured 20 m x 20 m describe in the plot descriptions. The canopy cover can be defined as the percent tree canopy area occupied by the vertical projection of tree crowns (Jennings, 1999). The percentage of canopy cover is measured by scathing the shadow of sunshine at ground level using 10 m x 10 m of white paper. The canopy projection when the sun was overhead was drawn to scale on white paper in each of four quadrants of the 20 m x 20 m plots, after which the areas shaded were cut out and weighed separately. Canopy cover was calculated according to eq. (5):

%Canopy Cover=(W Canopy)/(W Total) X 100 [5]

Where: %Canopy cover is the percentage of tree canopy cover, W_Canopy is the paper weight representing canopy cover and W_Total the paper weight representing the total area of observation, respectively.

For the reference we will considered as discussion part.

Comment 19 *20, section 2.5.2. Understorey and litter: reference to Hairiah et al. is missing from the reference list (presumably the CIFOR publication). Indicate the number of replications used please. Using 50 cm x 50 cm would seem inadequate for understory measurements in the case of Lantana or Chromolaena shrubs. Were these present in the forest plots like they are in many plantations across Java?

Feedback comment 19 Understorey vegetation and litter were measured according to the rapid carbon stock appraisal protocol (Hairiah et al., 2011), using 50 cm x 50 cm samples for fresh weight, with subsamples dried for dry weight determination. (Note: this method is standard for Carbon Stock measurement)

The reference used: Hairiah K, Ekadinata A, Sari RR, Rahayu S. Measurement of Carbon Stock: from land level to landscape. Practical instructions. Second edition. Bogor, World Agroforestry Centre, ICRAF SEA Regional Office, University of Brawijaya (UB), Malang, Indonesia 110

p.[Indonesia]http://apps.worldagroforestry.org/sea/Publications/files/manual/MN0049-11.pdf. 2011

Comment 20 *21, section 2.5.3. Surface roughness: awkward formulation ('elevation', 'vertically'). Suggest rephrasing.

Feedback comment 20 Measurement of difference of elevation is set with a pixel size of 30 cm x 30 cm. Each plot is divided into 6 pixels for 2 m meters width and 20 pixels for 6 m plot length, so there are 120 pixels (N). Pixels are made on a flat plane as high as 30 cm from the ground point of reference with a thin rope. In each center the pixel is measured vertically parallel to the thin rope vertically towards the surface of the ground with a ruler. The results of measurements of height differences in each pixel are used to calculate Ra with the equation (HoechstNetter et al., 2008):

$$Ra= 1/N \sum\nolimits_{(n=1)}^{N} h_n$$

Where N = Number of pixels in concerning patch; hn = difference of elevation between the n pixel in concerning patch and the mean value.

Comment 21 *22, line 187: daily rainfall totals do not represent rainfall 'intensity' although you might refer to 'event intensity' if event durations are known. Lines 188-190: this belongs to Discussion rather than Results and is rather speculative anyway given the non-linearity of the rainfall-erosion relationship. Add discussion on what might constitute 'tolerable soil loss' in the study area given the rate of chemical denudation of andesitic ashes (= approximate rate of soil formation; Verheijen et al., 2009). See e.g. Bruijnzeel (1983) who determined the rate of chemical weathering for a high rainfall area with Inceptisols in Central Java at ca. 85 t km-2 yr-1. Given the difference in rainfall between his site and the Rejoso catchment a value of ca. 40 t km-2 yr-1 might be defendable, suggesting the tolerable soil loss might be as low as 0.4 t ha-1 yr-1?

*23, lines 207-218 and 219-229: see main comments above.

Feedback comment 21 Erosion rates in all plots increased with amount of rainfall (Figure 5.a.3. and Figure 5.a.4). Midstream agroforestry systems had erosion rates range from 2.8 to 10.3 t ha-1 in the measurement period (Table 4). Move to discussion, inserted in line 230. As annual rainfall is approximately three times what was recorded in the measurement period, with similar rainfall intensities, these erosion rate are to be multiplied by a factor of 3, leading to 9 – 31 t ha-1 year-1. Even on volcanic soils, with frequent ash inputs, such erosion levels may be challenging sustainability.

Many authors have emphasized that the key to hydrologic functions is in the soil rather than the aboveground parts of the forest (Peña-Arancibia et al. 2019). Still, we found strong and direct relations with canopy cover. Positive effects of canopy cover on infiltration were related to raindrop interception in earlier studies (Carlesso et al. 2011; de Almeida et al. 2018). Interception will (a) reduce the destructive power of rainwater splash on the ground surface (as long as the effects Wiersum (1974) described are avoided, (b) allow more time for infiltration as water reaches the surface more slowly, (c) keep a thin water film on the leaves that will (d) cool the surrounding air when it subsequently evaporates. It will reduce the amount of water reaching the soil surface, but by increasing air humidity also decrease transpiration demand when stomata are open. In a study in North China, Li et al. (2014) showed that presence of litter of Quercus variabilis, representing broadleaf litter, and Pinus tabulaeformis, representing needle leaf litter, can reduce surface runoff rates by 29.5% and 31.3% respectively. The overall effect of fast plus slowly decomposing surface litter means protection of the soil surface from splash erosion, surface roughness that reduces sediment entrainment, an energy source for soil biota and a conducive microclimate (Hairiah et al. 2006, Derpsch et al., 2014)

Comment 22 *24, line 231: what was the original slope steepness in the mid-stream area before bench terracing? Line 232: awkward formulation. Line 233: remove reference to seawater intrusion since not pertinent to present case? Line 234: 'erodible', not 'erosive'. Line 240: Liu, not Zhipeng.

Feedback comment 22 Rev. line 231: From a land use policy perspective our results

suggest that maintaining high (∼80%) canopy cover in the mid-slope farmer-controlled landscape under bench terracing that does not match the slope criteria for designation as watershed protection forest, is important.

In Indonesia, protection forest areas have the primary function as protection of life support systems to regulate water management, prevent flooding, control soil erosion, and maintain soil fertility (Government of Indonesia, 1999).

With the higher rainfall intensities here and more erodible soils, risks for degradation from a downstream It clearly matters what the land cover in the 'non-forest' parts of the landscape is and how vegetation interacts with soils and geomorphology in shaping rivers and groundwater flows (Liu et al. 2018; Zhao et al. 2019).

Comment 23 *25, line 245: in the middle and upper Rejoso watershed; Line 247: keep erosion at acceptable levels? Line 248: gentle slopes associated with bench terracing or inherently gentle? If so, one wonders about the need for bench terracing.

Feedback comment 23 Rev: Our results demonstrated that vegetation-based thresholds for adequacy of infiltration, given the existing rainfall intensities, differed in the middle and upper Rejoso watershed.

Rev: Despite steep slopes and low tree cover, the upper watershed with its course soil texture (pseudo-sand /silt), low bulk density due to high content of amorf mineral, strong micro-aggregation and individual mineral have sponge-pores typical of Andosols, land management practices that combine vegetable crops with a tree canopy cover of around 55% can maintain infiltration and keep erosion at acceptable levels.

Rev: In the midstream part of the catchment, despite gentle slopes under bench terracing, infiltration-friendly land use on the fine-textured Inceptisols required a canopy cover of 80%.

Comment 24 *26, lines 251-253: remove 'was' (four times); Remove Didik Suprayogo in lines 252-253.

Feedback comment 24 We revised as: "DS, W, KH and MvN designed the study. NM collected data in midstream, ALR collected data in upstream, RMI coordinated to collect the data in the field, academically supervised by DS, W and KH. DS, W, KH and MvN shaped the manuscript, which was approved by all co-authors".

Comment 25 *27, references: standardize journal abbreviations, use of capitals, etc. Remove references not mentioned in text (Alvarenga, Anache, Boongaling, Choto, Kellner, Teklay); add missing references given in main text including Astutik et al. 2015; Hairiah et al. 2005, Hoechstetter et al. 2008, Suprayogo et al. 2017, etc.

Feedback comment 25 Corrected in revision text in supplement note 4

Comment 26 *28, diagrams: Figure 1, add latitude/longitude indications; Figure 2, move to Supporting Materials as it does not add much or replace by a Photo?; correct the volume of the collector drums; Figure 3, use less awkward dateindication; Figures 4, 5: indicate which panels refer to what land cover type; Figures 6-7: complete captions. Tables: add standard errors or coefficients of variation where appropriate.

Feedback comment 26 Regarding Tables we used Fisher's LSD test (p<0.05). Figure 3 we revised in supplement note 7.

Please also note the supplement to this comment:
https://www.hydrol-earth-syst-sci-discuss.net/hess-2020-2/hess-2020-2-AC1-supplement.pdf

**Supplement:**

**Supplement Revision note 1.**

Table 1. Land use, vegetation, soil conservation measure and slope of measurement plots

| Code | Land use | Vegetation (the average height of trees) | Terracing | Slope at plot level (%) |
|---|---|---|---|---|
| Upstream Rejoso watershed | | | | |
| UT1 | Old production forest | Pine (*Pinus merkusii*) (34 m) + grass | None | 35-40 |
| UT2 | Young production forest | Pine (11 m) + grass | None | 50-60 |
| UT3 | Agroforestry | Strip cemara (*Casuarina junghuniana)* (13 m) + Cabbage | None | 40-50 |
| UT4 | Arable land | Banana, maize, carrot | None | 40-50 |
| Midstream Rejoso watershed | | | | |
| MT1 | Old production forest | Mix Pine (28 m) or mahogany (*Swietenia macrophylla*) (12 m), banana, salak (*Salacca zalacca*), taro (*Colocasia esculenta*), elephant grass (*Miscanthus giganteus*). | Bench terrace sloping outward | 3-8 |
| MT2 | Agroforestry | Coffee-based (2 m) mix with durian (*Durio zibethinus*) (10 m), mahogany (9 m), *Leucaena leucocaphala* (8 m), *Paraserianthes falcataria* (11 m), *Albizia saman* (11 m), dadap (*Erythrina variegata*) (11 m), banana | Bench terrace sloping outward | 3-8 |
| MT3 | Agroforestry | Clove (*Syzygium aromaticum*) (8 m), banana | Bench terrace sloping outward | 3-8 |
| MT4 | Agroforestry | Manggo (*Mangifera indica*) (10 m), durian (10 m), *Randu kapuk* (*Ceiba pentandra*) (11 m) , maize, cassava, groundnut | Bench terrace sloping outward | 3-8 |

**Supplement Revision Note 2:**

[Figure]

: location of litter and understory measurements

Figure 1. The Scheme to measure the undergrowth or litter mass.

**Supplement Revision Note 3:** The additional data to revise Table 2.

[Figure]

**Figure: 2. Soil texture in five different layers in runoff plot measurements**

Table 2. bulk density,  particle density, soil porosity, macro-porosity  and organic C of runoff plots

a.  Upstream Rejoso watershed: Andisols

| Location code | Bulk Density (g cm⁻³)* | | | Particle Density (g cm⁻³)* | | | Soil porosity (%)* | | | Soil Macro-porosity (%) | | | $C_{org}$ (%)* | | |
|---|---|---|---|---|---|---|---|---|---|---|---|---|---|---|---|
| | At soil depth (cm) | | | | | | | | | | | | | | |
| | 0-10 | 10-20 | 20-30 | 0-10 | 10-20 | 20-30 | 0-10 | 10-20 | 20-30 | 0-10 | 10-20 | 20-30 | 0-10 | 10-20 | 20-30 |
| UT1 | 0.87a | 0.81a | 0.83a | 2.16a | 2.23a | 2.31a | 60a | 63a | 64c | 8.0b | 5.2b | 0.9a | 2.05bc | 1.61c | 1.79b |
| UT2 | 0.85a | 0.86a | 0.82a | 2.27a | 2.30a | 2.33a | 63a | 63a | 65c | 5.1ab | 1.5a | 0.3a | 2.46c | 1.56bc | 1.78b |
| UT3 | 0.81a | 0.84a | 0.85a | 2.14a | 2.12a | 2.28a | 62a | 60a | 63b | 4.7ab | 2.1ab | 1.4a | 1.17a | 0.58a | 0.71a |
| UT4 | 0.84a | 0.88a | 0.84a | 2.28a | 2.29a | 2.08a | 63a | 62a | 60a | 3.0a | 0.3a | 0.1a | 1.35ab | 1.06ab | 0.92a |
| LSD | 0.07 | 0.13 | 0.12 | 0.17 | 0.21 | 0.38 | 4 | 5 | 1 | 3.52 | 3.4 | 1.8 | 0.85 | 0.50 | 0.50 |

b.  Midstream Rejoso watershed: Inceptisols

| Location code | Bulk Density (g cm⁻³)* | | | Particle Density (g cm⁻³)* | | | Soil porosity (%)* | | | Soil Macro-porosity (%) | | | $C_{org}$ (%)* | | |
|---|---|---|---|---|---|---|---|---|---|---|---|---|---|---|---|
| | At soil depth (cm) | | | | | | | | | | | | | | |
| | 0-10 | 10-20 | 20-30 | 0-10 | 10-20 | 20-30 | 0-10 | 10-20 | 20-30 | 0-10 | 10-20 | 20-30 | 0-10 | 10-20 | 20-30 |
| MT1 | 0.83a | 0.85a | 0.83a | 2.20a | 2.28a | 2.20a | 62c | 63a | 62b | 13.6ab | 7.0bc | 2.5c | 1.73a | 1.87a | 1.65b |
| MT2 | 0.96b | 0.91a | 0.91a | 2.42b | 2.38a | 2.21a | 60bc | 62a | 59ab | 16.1b | 8.3c | 1.8bc | 2.22a | 1.59a | 1.84b |
| MT3 | 1.03bc | 0.96a | 0.94ab | 2.38b | 2.36a | 2.40a | 57ab | 59a | 61b | 11.7a | 3.4ab | 0.9ab | 2.19a | 1.61a | 1.01a |
| MT4 | 1.09c | 1.04a | 1.04b | 2.38b | 2.33a | 2.33a | 54a | 55a | 55a | 11.4a | 0.8a | 0 a | 1.71a | 1.36a | 1.12a |
| LSD | 0.10 | 0.24 | 0.11 | 0.15 | 0.17 | 0.22 | 4 | 10 | 4 | 4..0 | 3.9 | 1.0 | 0.84 | 0.54 | 0.41 |

*The same letter indicates no statistically significant differences between location with Fisher's LSD test (p<0.05).

Note: soil macro porosity measured using metyline blue method, will be describe in the Material and Method

[Figure]

Figure 3.  Soil Infiltration rate measured using double ring infiltrometer (n=6)

**Supplement Revision Note 4.**

[revised manuscript text omitted]

Climatic conditions that influence hydrology and erosion are largely determined by influence of the northwest and southwest monsoons. The northwest monsoon, picking up large amounts of moisture over the Indian Ocean, brings in most of the annual precipitation in the area, and predominates during the period from November through April. Although there is considerable variation in the amount and distribution of rainfall from year to year, most places in the watershed receive about four-fifth of the rainfall during the November-April wet season. Due to topographic influences, there is considerable spatial variation in annual precipitation as well, but generally ranges from 1500 mm to 3000 mm. The May to October period is considered the dry season. Then the southeast monsoon predominates, bringing much smaller amounts of precipitation due to the lower atmospheric moisture caused by lower temperatures in the Southern hemisphere at this time of the year. The rainfall distribution in upper-stream and Mid-stream is indicated that…..

Will be presenting graph monthly rainfall distribution from the average 10 years rain evens in Upper-stream and Mid-stream.

The Rejoso watershed watersheds consist of four types of soil, namely: Andisols, Inceptisols, Alfisols, and Entisols. Andisols are mainly distributed in the upperland, on the upper slopes of volcanoes. Andisols have a distinct black to very dark brown surface horizon within organic matter, which usually overlies a brown to dark yellowish brown subsoil. The clay fraction is dominated by allophane compounds. Andisols are highly permeable, porous with low bulk density, a high water-holding capacity and a crumb structure. The most common texture is sandy loam. Both soils have high inherent ferlity and are highly erodible only when seriously distributed. The midle and some lower volcanic slopes, consisting of easily weatherable permeable tuffs and ashes, give rise to deep stable soils – Inceptisols and Alfisols. Inceptisols are soils with only a limited horizon differenziation. Their texture ranges from deep friable clays to clay loams. Alfisols are soils which have accumulation of clay in the subsoil. Their texture ranges from loam to clay loam in the topsoil and clay loam to clay in the subsoil. Both soils have moderate to high inherent fertility but are highly susceptible to erosion. The Entisols are soils that lack horizon development and are found on volcanic sands, ashes and tuffs. Entisols occur on recent and sub-recent lahars of the Bromo volcanoes. Entisols with a coarse texture are extremely erodible and have very low water holding capacities. Permanent vegetative cover and especially diversified tree crops and agroforestry or forestry are most suitable land utilisation types to prevent erosion.

[Figure]

Figure 4.  The Rejoso Watershed from upstream (at the bottom) to sea level and land uses
considered to be a hydrological threat; purple indicates open soil, green tree cover.
(Modified from USGS, 2019).

**Supplement Revision  Note 6**

[Figure]

Figure 5.  Runoff and soil erosion plot measurements in  Rejoso Watershed.).

**Supplement Revision Note 7.**

[Figure]

Figure 6.  Distribution of rainfall during observation start on March 03, 2017 in the Rejoso watershed.

---

## Author Comment (AC2) · 17 May 2020

**1. General comment**

Comment 1 The study by Suprayogo et al. addresses a relevant topic and will be of interest to the hydrologic community. However, the work lacks the quality that is expected from a HESS publication. Some of the major issues are: 1) Introduction section is quite weak and doesn't offer enough lit review to gauge the novelty of the work; 2) research questions are vague and need to be carefully thought through; 3) Several method related details are unclear, and the statistical aspect of the work requires more effort. 4) Discussion lacks critical thinking, and authors need to carefully think about

the key messages that they want to discuss and convey. Lastly, the work needs clarity and organization throughout the sections. I would recommend the authors to revise the manuscript substantially and resubmit.

Feedback comment 1 Thank you, we do hope also that this manuscript is addressing a relevant topic to the hydrologic community. Our feedback with your concern: The Introduction, we will revised and the novelty we clarify at specific comment no 2.4. The research question, we develop hypothesis as describe at specific comment no. 2.17. The unclear method, we revised as drafted at specific comment from no 2.11, to no 2.17. The discussion, we revised as drafted as describe at specific comment no. 2.23.

Thank you for all the input for the improvement of this manuscript. The results of this review are really our concern to revise the manuscript.

Comment 2 Abstract – Too much unnecessary detail and the relevant information is missing. Most of the abstract focuses on introduction and methods and relatively little text on interpretations and implications of the work.

Feedback comment 2 Our abstract, we revised as: Abstract. Forest conversion to agriculture or agroforestry may increase risks of loss of hydrologic functions in an era of climate change. Infiltration during high-intensity rainfall is important for avoiding erosion and feeding aquifers but depends on land use practices that maintain soil macroporosity. In the forest-to-open-field-agriculture continuum it is not clear where thresholds to functionality ('degradation') are crossed. Our assessment of 'infiltration-friendly' land uses in the Rejoso watershed on the slopes of the Bromo volcano in East Java (Indonesia) focused on two zones, upstream (above 800 m a.s.l.) and midstream (400-800 m a.s.l.) of the Rejoso river and feeding aquifers that support lowland rice areas as well as drinking water supplies to nearby cities. Upstream land uses included old and young pine plantations (production forest) and highland vegetable crops with variation of tree canopy cover on the landscape on in steep (30-60%) to very steep (> 60%) lands with imperfect ridge terraces. Midstream land uses included production forest, multistrata

coffee-based agroforestry, clove-based agroforestry, and several mixed agroforestry types with variation of tree canopy cover on the landscape on moderately steep (15-30% and steep (30-60%) with bench terrace sloping outward. We quantified infiltration and erosion in three replications per land use category over a period of nearly two months (one-fourth of mean annual rainfall), with 6-13% of rainfall with intensities (51-100 mm day-1). We related soil infiltration to plot-level characteristics across the land use systems and found statistically significant relations with tree canopy cover (likely based on combined effects of interception, soil moisture status and effects on soil) and amount of litter. The soil infiltration had statistically significant relations with surface micro-topographic variation under forest-agroforestry systems (midstream), but not under vegetable cultivation in steep to very steep lands. There was no significant relationship between soil erosion and understory biomass of land use systems. Results for the upstream watershed showed that a tree canopy cover >55% was associated with adequate infiltration and acceptable soil erosion levels. For vegetable cultivation with a tree canopy cover below 55%, surface runoff was between 24% and 46% of rainfall, with high rates of soil erosion. Midstream, a tree canopy cover of >80% was associated with 'infiltration-friendly' land use, given the higher rainfall total (and rainfall intensity) in this zone. For a tree canopy cover in the range 20-80%, erosion rates were relatively low, but surface runoff increased to 36 to 62% of rainfall. Differences in soil type influenced the thresholds, as the areas' Inceptisols have lower intrinsic porosity than Andisols. A high soil surface roughness and litter thickness assist in reducing surface runoff and soil erosion. Where more open forms of agroforestry, with tree canopy cover less than 80% are becoming more common this will affect water resources in the downstream area and increase vulnerability to climate change.

Comment 3 Introduction – The authors need to highlight the novelty and research gap in their work. What is the research gap that you want to address? Is it about the volcanic landscape, erosion in a forest to open-field-agriculture continuum or, both? How many studies have done similar work in a volcanic landscape? If any, what were their findings?

Feedback comment 3 In the forest-to-open-field-agriculture continuum in the Rejoso watershed as examples, it is also happen in other place in Indonesia and also in the world, has the potential to alter water iňĆuxes on different spatial scales. Despite some large-scale studies being developed, there are still few investigations in experimental sites in this region. In Indonesia, many large-scale studies on water balance were developed in some hydrographic regions in the country (Ridwansyah et a;., 2014, Pradiko et al. 2015, Azmeri at al., 2015, Kuntoro et al., 2018, Mahmud et al. 2018, Nugroho et al. 2019). Nevertheless, experiment-scale studies are still rare due to the local heterogeneities and uncertainties from hydrological measurements and estimates (Beven, 2006; Graham et al., 2010).

Basic iňĄeld-hydrology studies are important for improving the sustainable land use management while promoting sustainable development. Therefore, these studies are important for promoting new solutions and techniques to maintain the water balance in spite of the rapid land use changes (Dotterweich, 2013; Nobrega et al., 2017). This research of this kind can be done using experimental plots, which are delimitated hillslopes (control volume) where the runoff is directed to one outlet of runoff colector (Sadeghi et al., 2013; Oliveira et al., 2016; Mwango et al., 2016; Strohmeier et al., 2016; Youlton et al., 2016b; Anache et al., 2017).

Comment 4 Research questions could use some clarity and can better. In question 1, ": : :limit iniňĄltration below the required rate" what do you mean by required rate? How do you define it? Please be more quantitative about it. Similarly, "iniňĄltration friendly" is also a vague term.

Feedback comment 4 Regarding the research question, we describe in general feed comment 7. Required rate is refer to adequate infiltration when the forest hydrological function with runoff coefficient is not more than 0.14 for Andisol (soil type A and for Inceptisol 0.20 (soil type C) base on reference: Knox County Tennessee. Stormwater Management Manual, section on the Rational Method, Volume 2 Technical Guidance. Available online: (https://dec.alaska.gov/water/wastewater/stormwater/swppp-
dev-runoff-coefficient-values-rm (accessed on 26 April 2020).

We considered this suggestion related the term "infįltration friendly", but think that it will create more confusion that it solves. If the reader gets interested in what this term means, we invite them to read the abstract and paper in which it is explained.

Comment 5 Generally, the topical sentence of a paragraph introduces an idea and rest of the texts expand on it. In the manuscript, several ideas have been introduced in the same paragraph, resulting in an incoherent passage with no clear message. This is true for many paragraphs throughout the draft. Lastly, the entire section could use better organization.

Feedback comment 5 Thank you for your suggestion, we will consider during revision of this manuscript

Comment 6 Methods Several method related details are unclear, and assumptions made during measurements have not been clearly laid out. How many places did you measure rainfall/through fall underneath the canopy? How do we account for the spatial variability in rainfall due to heterogeneity in canopy cover? The method used for estimating canopy cover is unclear, and the citation is missing in the References section.

Feedback comment 6 We clarify in the specific comment from comment 12 to comment 18.

Comment 7 "A probability level of 0.05 was set for rejecting null-hypothesis of no difference in tests of statistical.." what were the hypotheses that you tested? No hypothesis has been shared in the manuscript. Are you referring to LSD tests? In the fįeld conditions, most of the key drivers will "interact" in response to rainfall.

Feedback comment 7 To answer the first research question with the hypothesis that forest-to-open-field-agriculture continuum significantly decreases the soil hydrological function of forests, we examine differences in soil infiltration, runoff coefficient

and soil erosion between dominant land uses in each upstream and midstream with The Fisher's LSD test. The Fisher's LSD test, which establishes differences between groups defined for independent samples, was used for hypothesis testing, given that the data met the requirements for normality and homogeneity of variances. A probability level of 0.05 was set for rejecting null-hypotheses of no difference in tests of statistical significance. We used software GenStat 15th edition to have Fisher's LSD test. The soil infiltration, runoff coefficient and soil erosion then we compare with soil infiltration category (Landon, 1984), the adequate infiltration (Knox County Tennessee criteria) and acceptable soil erosion (calculated by using the formula Hammer, 1981). The second research question come out with hypothesis that dominant factors that determine "infiltration friendly" on plot scale are tree canopy cover, understorey vegetation, litter necromass, and land surface roughness. Linear regression relationships between the surface runoff / rainfall ratio or soil erosion and the amount of rainfall, tree canopy cover, understory, litter, and land surface roughness were determined using SigmaPlot version10.0.

The third research question is as an analysis the answers of the previous two research questions with the hypothesis that it is not always that the upstream watershed area is more sensitive to hydrological disturbance due to changes in land use than midstream, but the factor of soil properties also determines considerations in watershed hydrological management.

Comment 8 The regression model with a single explanatory variable may not be an ideal approach here. Would it be possible to use models that incorporate some interaction effect between the explanatory variables? Also, did you explore the correlation among explanatory variables? If you are really interested in exploring the influence of an individual exploratory variable on erosion, please use partial correlations that allows to control the effect of other related variables on responses.

Feedback comment 8 Thank you for your suggestion, We will looking again the data and we try to reanalysis by using the linier multiple regression with Genstat statistical

tool analysis. However, we will still consider with 12 observations whether it is sufficient for this analysis.

Comment 9 Results and Discussion: The discussion section could use some critical thinking, especially about the key messages that you want to convey. Most of the discussion revolves around the major drivers of responses. The section barely discusses questions 1 and 3. Lastly, you want to show how the work advances our current understanding of runoff and erosion processes in the volcanic region.

Feedback comment 9 The result and discussion, we will revised to answers the determined tree research question in this manuscript. For the first research question, we added the additional data that not yet presented in the first draft of manuscript. The additional data as presented in supplement note no 1. as revised Table 2. We revised soil texture data that presented on each layering soil of 0-10 cm, 10-20 cm and 20-30 cm, 30-40 and 40-50 cm. We added our data measurement on soil bulk density, particle density, total soil porosity, soil macro-poroity and soil organic matter content in each layering soil of 0-10 cm, 10-20 cm and 20-30 cm. We also will added the infiltration measurement data by using Double ring infiltrometer as presented in note 1.2. We use this data to answer the first research question.

2. SpeciïñĄc comments

Comment 1 Line 11 Too much detail and the key massages are hard to follow

Feedback comment 1 We clarify in the general comment 2 above.

Comment 2 Line 24.What was the landscape type steep, very steep? Why mention at one place not at others?

Feedback comment 2 We classified based on FAO, 2006. Guidelines for soil description. Food and Agriculture Organization of the United Nations, Rome, 2006.

We revised: Upstream land uses included old and young pine plantations (production forest) and highland vegetable crops with variation of tree canopy cover on the landscape on in steep (30-60%) to very steep (> 60%) lands with imperfect ridge terraces. Midstream land uses included production forest, multistrata coffee-based agroforestry, clove-based agroforestry, and several mixed agroforestry types with variation of tree canopy cover on the landscape on moderately steep (15-30% and steep (30-60%) with bench terrace sloping outward.

Comment 3 Line 24. How do you define adequate infiltration or acceptable soil erosion? Please be more quantitative

Feedback comment 3 Adequate infiltration when the forest hydrological function with runoff coefficient is not more than 0.14 for Andisol (soil type A and for Inceptisol 0.20 (soil type C) base on reference: Knox County Tennessee. Stormwater Management Manual, section on the Rational Method, Volume 2 Technical Guidance. Available online: (https://dec.alaska.gov/water/wastewater/stormwater/swppp-dev-runoff-coefficient-values-rm (accessed on 26 April 2020).

Acceptable soil erosion where the soil erosion equal or less than Agriculture permissible rate of soil erosion (Eapr, ton ha-1 year-1) for tropical soil analyzed according to the formula Hammer [1981], calculated as::

E_apr=((Depth of soil*Factor of soil depth)/(The useful of soil))*Soil bulk density

Both Andisol and Inceptisol is has deep soil, then we consider that the regolith soil has 120 cm soil depth, and factor of soil depth = 1 (see also supplement note no 2), The useful of soil we determine 400 year. With average soil bulk density of Andisol = 0.83 g cm-3, and Inceptisol = 0.99 g cm-3, then the Eapr Andisol = 24.9 ton ha-1 year-1 and Eapr Inceptisol 29.7 ton ha-1 year-1

Hammer W. I. 1981 Soil Conservation Consultant Report AGOF INS/78/006 Technical Note No 7CSR Bogor

Comment 4 Line 29 Threshold for what?

Feedback comment 4 We mean that threshold for "infiltration friendly"

Comment 5 Line 33 Research gap and novelty are difficult to see here

Feedback comment 5 In the forest-to-open-field-agriculture continuum in the Rejoso watershed as examples, it is also happen in other place in Indonesia and also in the world, has the potential to alter water fluxes on different spatial scales. Despite some large-scale studies being developed, there are still few investigations in experimental sites in this region. In Indonesia, many large-scale studies on water balance were developed in some hydrographic regions in the country (Ridwansyah et a;., 2014, Pradiko et al. 2015, Azmeri at al., 2015, Kuntoro et al., 2018, Mahmud et al. 2018, Nugroho et al. 2019). Nevertheless, experiment-scale studies are still rare due to the local heterogeneities and uncertainties from hydrological measurements and estimates (Beven, 2006; Graham et al., 2010). Basic field-hydrology studies are important for improving the sustainable land use management while promoting sustainable development. Therefore, these studies are important for promoting new solutions and techniques to maintain the water balance in spite of the rapid land use changes (Dotterweich, 2013; Nobrega et al., 2017). This research of this kind can be done using experimental plots, which are delimitated hillslopes (control volume) where the runoff is directed to one outlet of runoff colector (Sadeghi et al., 2013; Oliveira et al., 2016; Mwango et al., 2016; Strohmeier et al., 2016; Youlton et al., 2016b; Anache et al., 2017). References listed at supplement note no 3.

Comment 6 Line 37 Almost every line in the paragraph introduction a new idea and moves on to other in the next line. What is the massage that you want to convey here?

Feedback comment 6 Thank you for your suggestion, we will consider during revision of this manuscript

Comment 7 Line 71 How do we know this? Any citation / data?

Feedback comment 7 This is based on the video document made by Danon and we will share this video

Comment 8 Line 79 What do you mean by required rate?

Feedback comment 8 Required rate is refer to adequate infiltration when the forest hydrological function with runoff coefficient is not more than 0.14 for Andisol (soil type A and for Inceptisol 0.20 (soil type C) base on reference: Knox County Tennessee. Storm-water Management Manual, section on the Rational Method, Volume 2 Technical Guidance. Available online: (https://dec.alaska.gov/water/wastewater/stormwater/swppp-dev-runoff-coefficient-values-rm (accessed on 26 April 2020).

Comment 9 Line 80 Discussion sector mostly focuses on the second question, rest of questions are rarely discussed

Feedback comment 9 Thank you for your suggestion, we will revised the balance discussion for three research question in this manuscript

Comment 10 Line 85 Please use SI units throughout

Feedback comment 10 63,359 hectares. We revised using SI unit become 634 km2

Comment 11 Line 100 This is also known as runoff coefficient

Feedback comment 11 We will change in the manuscript instead of the Runoff / Rainfall ratio to become runoff coefficient.

Comment 12 Line 101 Did you measure rainfall at multiple places under the canopy or at 'one' places? Generally, there is large variability an interception, depending upon the vegetation / forest type

Feedback comment 12 We revised as: "Rain gauges were installed in four observation locations (with adjacent erosion plots) upstream and four observation locations midstream of the Rejoso watershed. Each plot, the rainfall was measured with 5 replications. The rain gauges consisted of 30 cm diameter of funnel and bottle with a volume of 1.5 dm3 placed 120 cm above the soil surface and below the tree canopy

with bamboo as a support. Rainfall was observed every day during two months of the rainy season, from March to May 2017".

The measurement of rainfall at 5 places randomly in the inner-plot. It is right there is large variability an interception, depending upon the vegetation / forest type. The Through-fall /Rainfall ratio variability is presented in supplement note no 4.

Comment 13 Line 112 What do you mean by "newsprint"? Is that standard approached? Please provided citation.

Feedback comment 13 "Newsprint" mean old newspapers (see picture in Supplement Note 5). We filtering the sediment sample using 'newsprint'. The water through a filter 'newsprint' was relatively clear. Therefore, we consider that 'newsprint' filter is considered effective to trap the sediment. We do not calibrate between 'newsprint' filter with standard filter. However, we can calibrate this filter 'newsprint' with standard filter.

Comment 14 Line 131. I was expecting remote sensing based method here. I can't find citation so don't really know about this method.

Feedback comment 14 The canopy cover can be defined as the percent tree canopy area occupied by the vertical projection of tree crowns (Jennings, 1999). The percentage of canopy cover is measured by scathing the shadow of sunshine at ground level using 10 m x 10 m of white paper. The canopy projection when the sun was overhead was drawn to scale on white paper in each of four quadrants of the 20 m x 20 m plots, after which the areas shaded were cut out and weighed separately. Canopy cover was calculated according to eq. (5):

$$\%Canopy\ Cover = (W\ Canopy)/(W\ Total) \times 100 \quad [5]$$

Where: % Canopy cover is the percentage of tree canopy cover, W_Canopy is the paper weight representing canopy cover and W_Total the paper weight representing the total area of observation, respectively. *Jennings, S.B., N.D. Brown, and D. Sheil..

Assessing forest canopies and understorey illumination: Canopy closure, canopy cover and other measures. Forestry72(1):59-73. 1999.

Comment 15 Line 133 Citation not provided

Feedback comment 15 Cited Arumsari, 2003 is in Bahasa Indonesia and hence less accessible for most readers therefore, we replace with *Jennings, S.B., N.D. Brown, and D. Sheil.. Assessing forest canopies and understorey illumination: Canopy closure, canopy cover and other measures. Forestry72(1):59-73. 1999.

Comment 16 Line 137. CV also refer to coefficient of variation so may bay pick a different acronym

Feedback comment 16 We change without acronym as below:

Canopy cover was calculated according to eq. (5): %Canopy Cover=(W Canopy)/(W Total) X 100 [5]

Where: % Canopy cover is the percentage of tree canopy cover, W_Canopy is the paper weight representing canopy cover and W_Total the paper weight representing the total area of observation, respectively

Comment 17 Line 144 Again citation missing in reference.

Feedback comment 17 Hoechsteetter, S., Walz, U., Dang, L. H., Thinh, N. X.. Effect of Topography and Surface Roughness in Analyses of Landscape Structure- A Proposal Modify The Existing Set of Landscape Metrics. Landscape Online 3: 1-14 (8). https://gfzpublic.gfz-potsdam.de/pubman/item/item_32854. 2008

Comment 18 Line 145. Provide more detail. What are the groups that you are looking at. What hypotheses are being tested? Are you planning to build linier regression model to stimulate runoff or sediment? Line 147 I haven't seen any hypothesis yet?

Feedback comment 18 To answer the first research question with the hypothesis that forest-to-open-field-agriculture continuum significantly decreases the soil hydrological

function of forests, we examine differences in soil infiltration, runoff coefficient and soil erosion between dominant land uses in each upstream and midstream with The Fisher's LSD test. The Fisher's LSD test, which establishes differences between groups defined for independent samples, was used for hypothesis testing, given that the data met the requirements for normality and homogeneity of variances. A probability level of 0.05 was set for rejecting null-hypotheses of no difference in tests of statistical significance. We used software GenStat 15th edition to have Fisher's LSD test. The soil infiltration, runoff coefficient and soil erosion then we compare with soil infiltration category (Landon, 1984), the adequate infiltration (Knox County Tennessee criteria) and acceptable soil erosion (calculated by using the formula Hammer, 1981). The second research question come out with hypothesis that dominant factors that determine "infiltration friendly" on plot scale are tree canopy cover, understorey vegetation, litter necromass, and land surface roughness. Linear regression relationships between the surface runoff / rainfall ratio or soil erosion and the amount of rainfall, tree canopy cover, understory, litter, and land surface roughness were determined using SigmaPlot version10.0.

The third research question is as an analysis the answers of the previous two research questions with the hypothesis that it is not always that the upstream watershed area is more sensitive to hydrological disturbance due to changes in land use than midstream, but the factor of soil properties also determines considerations in watershed hydrological management.

Comment 19 Line 155. What do mean by average rainfall here? This is the rainfall total for 3 months correct? How did you get range, explain!; rewrite the entire sentence for clarity

Feedback comment 19 Yes, it is right that the rainfall total of 2 months (not 3 months). The average rainfall mean that in upstream we have 4 condition of canopy cover and each canopy cover has 3 replication, then there is 12 plot measurements. Each plot of measurement was installed a rain gouge on open surface. We also installed five rain

gouges to measure through-fall rain. This rainfall is the average of 12 rain gouge on open surface, then we have a range. The same thing also done for midstream.

Revision: Within the measurement period 31 rainy days were recorded (Figure 3). Rainfall variation between upstream and midstream observation plots was relatively high with an average of 520 mm (range 476 - 556 mm among 12 rain gauge measurements), and an average 666 mm (range 541 – 840 mm among 12 rain gauge measurements), respectively. In the upstream and midstream areas 71% and 57% of the rainy days had <20 mm day-1 ('light rain'), 24% and 31% had 'moderate' rainfall (21-50 mm day-1) and 6% and 13% 'heavy' rain (51-100 mm day-1), respectively; none had 'very heavy rain' (> 100 mm day-1). Such rain conditions indicate that the rain-erosivity in midstream is higher than that upstream.

Comment 20 Line 163 what is C-org? never been explained in the paper. Organic carbon content?

Feedback comment 20 We revised become: The soil organic carbon content varied from 0.65 to 2.12 %.

Comment 21 Line 197 It is correct interception? Figure 6 and Figure 7 do not show no significant relationships with understory, no?

Feedback comment 21 Thank you for your correction. We revised as: "Understory vegetation theoretically can reduces splash impacts on the soil and supports infiltration, as does the litter necromass present. However, the result of this study indicated that understory show no significant relationships with runoff coefficient and soil erosion".

Comment 22 Line 203 Is this the summary of all the findings?

Feedback comment 22 Referring the comment 2.17 this is the summary of the finding for research question 2. We will add the summary research question 1 and 3.

Comment 23 Line 207 Itn't this broad statement? Won't also depend upon the type of forest, no?

Feedback comment 23 We revised as drafted in Supplement Note 6.

Comment 24 Line 207 para 2 on canopy cover, para 3-roughness and porosity

Feedback comment 24 We revised as drafted in Supplement Note 6.

Comment 25 Line 219 Bit vague statement, compared to?

Feedback comment 25 We revised as "Both the production forest and agroforestry systems with high canopy maintained a relatively high land surface roughness compared with rare canopies in midstream area."

Comment 26 Line 220 Refer to?

Feedback comment 26 We revised as "Without a high canopy cover (Table 3.a). this roughness was not able to control surface runoff and erosion in. . .."

Comment 27 Line 221 Significant correlation with what?

Feedback comment 27 We revised as ". . .. . .this roughness was not able to control surface runoff and erosion in the upstream area, but midstream there were significant correlations with runoff coefficient and soil erosion".

Comment 28 Line 221 I understand but it would be better to elaborate and explain clearly

Feedback comment 28 We revised as "The role of surface roughness as sediment filter may depend on frequent regeneration to counter homogenisation (Rodenburg et al. 2003). Surface roughness in the landscape includes a cavity, meandering of streams due to the present of litter, necromass, tree trunk and rocks, providing opportunities for water flow to stop for longer periods and experience infiltration. This condition also functions as a sediment filter. This function needs to be managed through land management, so that surface roughness is maintained on the ground".

Comment 29 Line 222 Quite odd! Why sudden switch from surface roughness to porosity in the same paragraph? Shouldn't porosity be the part of soil related discussion?

Feedback comment 29 We revised as drafted in Supplement Note 6.

Comment 30 Line 225 broad, please be spesific

Feedback comment 30 We revised as "The formation of old tree root channels can cause long time-lags between land cover change and soil macroporosity (van Noordwijk et al. 2011), obscuring relations between current tree cover and soil hydrologic functions"

Comment 31 Line 247 How do we define acceptable? Any value /range of infiltration rate?

Feedback comment 31 Required rate is refer to adequate infiltration when the forest hydrological function with runoff coefficient is not more than 0.14 for Andisol (soil type A and for Inceptisol 0.20 (soil type C) base on reference: Knox County Tennessee. Storm-water Management Manual, section on the Rational Method, Volume 2 Technical Guidance. Available online: (https://dec.alaska.gov/water/wastewater/stormwater/swppp-dev-runoff-coefficient-values-rm (accessed on 26 April 2020).

Comment 32 Line 379 you may want to say this is a "plan view"

Feedback comment 32 We revised "Figure 2. Runoff and soil erosion plot"

Comment 33 Line 385 Please briefly remind readers about what "1-4" refers to

Feedback comment 33 We rivesed as "Figure 5: Soil erosion in relation to daily rainfall rates in production forest and agroforestry (a) Upstream Rejoso Watershed, under (a.1) 55% canopy cover of Pine based of old production forest, (a.2) 40% canopy cover of Pine based of young production forest, (a.3) 5% canopy cover of Cemara based of Agroforestry with Cabbage crop, (a.4) 0% tree canopy cover of Arable land (maize crop) ; (b) Midstream Rejoso Watershed under (b.1) 87 % canopy cover of Pine/ mahogany

based of old production forest, (b.2) 75% canopy cover of Coffee-based agroforestry, (b.3) 52 % canopy cover of Clove based agroforestry, (b.4) 26% canopy cover of mix tees-crop based agroforestry".

Comment 34 Figure 6. Ideally, these dirvers / variables interact which these linear regression models are not capturing. I am not sure if you have enough sample size to build a bit more sophisticated model that came in corporates interaction effect

Feedback comment 34 We have tried multiple linear regression analysis but the relationship is not good. This is indeed true that with n = 12 there is insufficient data to get an acceptable statistical analysis. For this reason, we believe that it should still be presented as presented in the manuscript.

Comment 35 Why not lenear relationship in upstream and liner in mid-stream- any hypothesis?

Feedback comment 35 Figure 6 we present following the distribution of data trends, then we determine the relationships that best fit with existing data trends. The hypothesis is the same that is "dominant factors that determine "infiltration friendly" on plot scale are tree canopy cover, understorey vegetation, litter necromass, and land surface roughness"

Comment 36 No relation with understory?

Feedback comment 36 This is possible because surface runoff and erosion have been largely controlled by land cover. Growth and development of understory determined by canopy cover. Likewise, the tree plantations in each plot are also diverse, so this also affects the diversity of understorey vegetation underneath.

Comment 37 Figure 7. Add n sample size to the plot or caption

Feedback comment 37 Figure 7. Soil erosion in relation to tree canopy cover, understorey vegetation, litter necromass, and land surface roughness (n=12) in a: Upstream Rejoso Watershed; and b: Midstream Rejoso Watershed.

**Supplement:**

**Supplement Feedback to comments by anonymous Referee of**

**"Infiltration-Friendly Land Uses for Climate Resilience on Volcanic Slopes in the Rejoso Watershed, East Java, Indonesia"**

**by Didik Suprayogo et al.**

**Supplement Note 1:** The additional data to revise Table 2.

[Figure]

Figure 1. Soil texture in five different layers in runoff plot measurements

Table 2. bulk density,  particle density, soil porosity, macro-porosity  and organic C of runoff plots

a. Upstream Rejoso watershed: Andisols

| Location code | Bulk Density (g cm⁻³)* | | | Particle Density (g cm⁻³)* | | | Soil porosity (%)* | | | Soil Macro-porosity (%) | | | $C_{org}$ (%)* | | |
|---|---|---|---|---|---|---|---|---|---|---|---|---|---|---|---|
| | At soil depth (cm) | | | | | | | | | | | | | | |
| | 0-10 | 10-20 | 20-30 | 0-10 | 10-20 | 20-30 | 0-10 | 10-20 | 20-30 | 0-10 | 10-20 | 20-30 | 0-10 | 10-20 | 20-30 |
| UT1 | 0.87a | 0.81a | 0.83a | 2.16a | 2.23a | 2.31a | 60a | 63a | 64c | 8.0b | 5.2b | 0.9a | 2.05bc | 1.61c | 1.79b |
| UT2 | 0.85a | 0.86a | 0.82a | 2.27a | 2.30a | 2.33a | 63a | 63a | 65c | 5.1ab | 1.5a | 0.3a | 2.46c | 1.56bc | 1.78b |
| UT3 | 0.81a | 0.84a | 0.85a | 2.14a | 2.12a | 2.28a | 62a | 60a | 63b | 4.7ab | 2.1ab | 1.4a | 1.17a | 0.58a | 0.71a |
| UT4 | 0.84a | 0.88a | 0.84a | 2.28a | 2.29a | 2.08a | 63a | 62a | 60a | 3.0a | 0.3a | 0.1a | 1.35ab | 1.06ab | 0.92a |
| LSD | 0.07 | 0.13 | 0.12 | 0.17 | 0.21 | 0.38 | 4 | 5 | 1 | 3.52 | 3.4 | 1.8 | 0.85 | 0.50 | 0.50 |

b. Midstream Rejoso watershed: Inceptisols

| Location code | Bulk Density (g cm⁻³)* | | | Particle Density (g cm⁻³)* | | | Soil porosity (%)* | | | Soil Macro-porosity (%) | | | $C_{org}$ (%)* | | |
|---|---|---|---|---|---|---|---|---|---|---|---|---|---|---|---|
| | At soil depth (cm) | | | | | | | | | | | | | | |
| | 0-10 | 10-20 | 20-30 | 0-10 | 10-20 | 20-30 | 0-10 | 10-20 | 20-30 | 0-10 | 10-20 | 20-30 | 0-10 | 10-20 | 20-30 |
| MT1 | 0.83a | 0.85a | 0.83a | 2.20a | 2.28a | 2.20a | 62c | 63a | 62b | 13.6ab | 7.0bc | 2.5c | 1.73a | 1.87a | 1.65b |
| MT2 | 0.96b | 0.91a | 0.91a | 2.42b | 2.38a | 2.21a | 60bc | 62a | 59ab | 16.1b | 8.3c | 1.8bc | 2.22a | 1.59a | 1.84b |
| MT3 | 1.03bc | 0.96a | 0.94ab | 2.38b | 2.36a | 2.40a | 57ab | 59a | 61b | 11.7a | 3.4ab | 0.9ab | 2.19a | 1.61a | 1.01a |
| MT4 | 1.09c | 1.04a | 1.04b | 2.38b | 2.33a | 2.33a | 54a | 55a | 55a | 11.4a | 0.8a | 0 a | 1.71a | 1.36a | 1.12a |
| LSD | 0.10 | 0.24 | 0.11 | 0.15 | 0.17 | 0.22 | 4 | 10 | 4 | 4..0 | 3.9 | 1.0 | 0.84 | 0.54 | 0.41 |

*The same letter indicates no statistically significant differences between location with Fisher's LSD test (p<0.05).

Note: soil macro porosity measured using metyline blue method, will be describe in the Material and Method

[Figure]

Figure 2. Soil Infiltration rate measured using double ring infiltrometer (n=6)

**Supplement Note 2**

Table : The depth factor of some soils in Indonesia (Hammer, 1981)

| Soil taxonomy | Soil degradation | | Factor of soil depth |
|---|---|---|---|
| Sub order | Physical | Chemical | |
| Aqualf (AQ) | M | L | 0.90 |
| Udalf (AD) | M | L | 0.90 |
| Andept (IN) | L | L | 1.00 |
| Aquept (IQ) | L | M | 0.95 |
| Tropept (IT) | L | L | 1.00 |
| Udult (UD) | M | M | 0.80 |

**Supplement Note 3**

Anache, J. A. A., Wendland, E. C., Oliveira, P. T. S., Flanagan, D. C., and Nearing, M. A.: Runoff and soil erosion plot-scale studies under natural rainfall: A metaanalysis of the Brazilian experience, Catena, 152, 29–39, https://doi.org/10.1016/j.catena.2017.01.003, 2017.

Azmeri, Yulianur A. , Rizalihadi M., and Bachtiar, S. Hydrological Response Unit Analysis Using AVSWAT 2000 for Keuliling Reservoir Watershed, Aceh Province, Indonesia. Aceh Int. J. Sci. Technol., 4(1): 32-40. 2015.

Beven, K.: On undermining the science?, Hydrol. Process., 20, 3141–3146, https://doi.org/10.1002/hyp.6396, 2006.

Dotterweich, M.: The history of human-induced soil erosion: Geomorphic legacies, early descriptions and research, and the development of soil conservation – A global synopsis, Geomorphology, 201, 1–34, https://doi.org/10.1016/j.geomorph.2013.07.021, 2013.

Graham, C. B., van Verseveld, W., Barnard, H. R., and McDonnell, J. J.: Estimating the deep seepage component of the hillslope and catchment water balance within a measurement uncertainty framework, Hydrol. Process., 24, 3631–3647, https://doi.org/10.1002/hyp.7788, 2010.

Kuntoro A.A., Cahyono M. and Soentoro E.A. Land Cover and Climate Change Impact on River Discharge: Case Study of Upper Citarum River Basin.. J. Eng. Technol. Sci., Vol. 50, No. 3: 364-381. http://journals.itb.ac.id/index.php/jets/article/view/8557 2018.

Mahmud , Kusumandari A. , Sudarmadji , and Supriyatno N. A Study of Flood Causal Priority in Arui Watershed, Manokwari Regency, Indonesia. Jurnal Manajemen Hutan Tropika Vol. 24, (2): 81-94. https://journal.ipb.ac.id/index.php/jmht/article/view/21380/16321 2018

Mwango, S. B., Msanya, B. M., Mtakwa, P. W., Kimaro, D. N., Deckers, J., and Poesen, J.: Effectiveness OF Mulching Under Mirabain Controlling Soil Erosion, Fertility Restoration and Crop Yield in the Usambara Mountains, Tanzania, Land Degrad. Dev., 27, 1266–1275, https://doi.org/10.1002/ldr.2332, 2016.

Nobrega, R. L. B., Guzha, A. C., Torres, G. N., Kovacs, K., Lamparter, G., Amorim, R. S. S., Couto, E., and Gerold, G.: Effects of conversion of native cerrado vegetation to pasture on soil hydro-physical properties, evapotranspiration and streamflow on the Amazonian agricultural frontier, Plos One, 12, e0179414, https://doi.org/10.1371/journal.pone.0179414, 2017.

Nugroho S.P., Handayani L.D.W, Meidiza R. and Munggaran. G. Land use change analysis for hydrology response and planning management of Cibeet Sub-Watershed,West Java, Indonesia

IOP Conf. Series: Earth and Environmental Science **284**: 012002 IOP Publishing doi:10.1088/1755-1315/284/1/012002 https://iopscience.iop.org/article/10.1088/1755-1315/284/1/012002/pdf. 2019.

Oliveira, P. T. S., Nearing, M. A., Hawkins, R. H., Stone, J. J., Rodrigues, D. B. B., Panachuki, E., and Wendland, E.: Curve number estimation from Brazilian Cerrado rainfall and runoff data, J. Soil Water Conserv., 71, 420–429, https://doi.org/10.2489/jswc.71.5.420, 2016

Pradiko H., Arwin, Soewondo P., Suryadi Y. The change of hydrological regime in upper Cikapundung Watershed, West Java Indonesia. The 5th International Conference of Euro Asia Civil Engineering Forum (EACEF-5). Procedia Engineering 125: 229 – 235. https://reader.elsevier.com/reader/sd/pii/S1877705815033500 2015.

Ridwansyah, I., Pawitan H., Sinukaban N.. and Hidayat, Y. Watershed Modeling with ArcSWAT and SUFI2In Cisadane Catchment Area: Calibration and Validation to Prediction of River Flow. International Journal of Science and Engineering, Vol. 6(2):12-21,. https://ejournal.undip.ac.id/index.php/ijse/article/view/5975/pdf 2014.

Sadeghi, S. H. R., Seghaleh, M. B., and Rangavar, A. S.: Plot sizes dependency of runoff and sediment yield estimates from a small watershed, Catena, 102, 55–61, https://doi.org/10.1016/j.catena.2011.01.003, 2013.

Strohmeier, S., Laaha, G., Holzmann, H., and Klik, A.: Magnitude and Occurrence Probability of Soil Loss: A Risk Analytical Approach for the Plot Scale For Two Sites in Lower Austria, Land Degrad. Dev., 27, 43–51, https://doi.org/10.1002/ldr.2354, 2016.

Youlton, C., Wendland, E., Anache, J. A. A., Poblete-Echeverría, C., and Dabney, S.: Changes in erosion and runoff due to replacement of pasture land with sugarcane crops, Sustainability-Basel, 8, 685, https://doi.org/10.3390/su8070685, 2016.

**Supplement Note 4. Through-fall /Rainfall ratio variability**

[Figure]

Figure.. The Through-fall /Rainfall ratio variability in measured runoff plot, Rejoso Watershed

**Supplement Note 5.** Filtering sediment using old newspapers

[Figure]

[Figure]

[Figure]

**4 Discussion**

The first research question with hypothesis that forest-to-open-field-agriculture continuum significantly decreases the soil hydrological function of forest. The results of the present study confirms that land use type from high density of trees in forest were significantly decreased soil infiltration rate compared with low density of trees in agroforestry (Table . 2). The results of this study are also supported from the results of previous studies, where found that decreases in ground cover often resulted in decreases in soil infiltration rate (Gifford and Hawkins, 1978, Suprayogo et al., 2004, Forests have been shown to reduce surface runoff and erosion compared with coffee monoculture (Widianto et al., 2004). Neris et al. (2012) also found that soil infiltration in both green forest and pine forest higher than cropland. Ma et al. (2007) who found that soil infiltration rate of forest was greater than that of agroforestry. Sun et al., (2018) reported from their meta-analysis that converting any land use type with vegetation cover to crop land is in favour of decline of soil infiltration rate and are not beneficial to soil and water conservation. However, Wang et al. (2015) reported that conversion from forest to agroforestry increased soil infiltration rate.

The degradation of soil hydrological function of forest could be attributed to the decrease of soils' macroporosity and organic matter content, increased soil bulk density (Fig…..) which had significantly positive correlation with soil initial and steady infiltration rates (Fig. ….). Among various land use patterns, plant root activities are important factors affecting soil infiltration (Butt and Bowman, 2002; Zimmermann et al., 2006). The reason why cropland has lower infiltration rate than The and use type from high density of trees compared with low density of trees in forest may be verified by the fact that soils beneath the canopies of woody plants had a more extensive distribution of plant roots and a greater number of macropores, biologically-produced pores (Dunkerley, 2000; Colloff et al., 2010), which created a positive feedback on infiltration (Reid et al., 1999; Bhark and Small, 2003). The soils' macroporosity, needed for effective infiltration, is the result of a continuous process of compaction and filling in of macropores with fine soil particles, and creation of biogenic channels (formed by old tree roots, earthworms and other soil engineers) or abiotic processes (cracks). As no heavy machinery is used in any of these land use systems, compaction is restricted to human feet, and motorbikes in specific tracks. The formation of old tree root channels can cause long time-lags between land cover change and soil macroporosity (van Noordwijk et al. 2011), obscuring relations between current tree cover and soil hydrologic functions. Zwartendijk et al. (2017) showed that 'fallows' were intermediate between forests and grasslands in terms of infiltration in Madagascar. Recovery of infiltration after reforestation of grasslands in the Philippines was found to be a matter of decades rather than years (Zhang et al. 2019). Suprayogo et al. (2004) proves that forest soils have relatively more pore macro and higher surface infiltration rates than monoculture coffee plantations. Land use change, especially from forest to cropland, have caused remarkable changes in soil properties including loss of organic matter and increases in bulk density (Lepsch et al., 2010), which lead to decrease infiltration rate (Mwendera and Saleem, 2010). Some researchers suggested a positive relationship between soil organic matter and infiltration rate (Martens and Frankenberger, 1992; Osuji et al., 2010).

The second research question come out with hypothesis that dominant factors that determine "infiltration friendly" on plot scale are tree canopy cover, understorey vegetation, litter necromass, and land surface roughness. Our research show that a number of land cover types had infiltration rates below the required rates at peak rainfall events. Among the four factors tested, tree cover and litter layer necromass could be used to define zone-specific thresholds for infiltration-friendly land use, but understory vegetation and surface roughness did not. Although slopes in the upper watershed are much steeper than midstream, the coarser texture and likely higher aggregate stability means that thresholds for canopy cover and litter necromass can be lower.

Many authors have emphasized that the key to hydrologic functions is in the soil rather than the aboveground parts of the forest (Peña-Arancibia et al. 2019). Still, we found strong and direct relations with canopy cover. Positive effects of canopy cover on infiltration were related to raindrop interception in earlier studies (Carlesso et al. 2011; de Almeida et al. 2018). Interception will (a) reduce the destructive power of rainwater splash on the ground surface (as long as the effects Wiersum (1974) described are avoided, (b) allow more time for infiltration as water reaches the surface more slowly, (c) keep a thin water film on the leaves that will (d) cool the surrounding air when it subsequently evaporates. It will reduce the amount of water reaching the soil surface, but by increasing air humidity also decrease transpiration demand when stomata are open.

Understory vegetation theoretically can reduces splash impacts on the soil and supports infiltration, as does the litter necromass present. However, the result of this study indicated that understory show no significant relationships with runoff coefficient and soil erosion. This is possible because surface runoff and erosion have been largely controlled by land cover. Growth and development of understory determined by canopy cover. Likewise, the tree plantations in each plot are also diverse, so this also affects the diversity of understorey vegetation underneath.

The litter function is provided by thick litter closure at the ground surface. The result of this study indicate that litter layer in the old production forest both in upstream and midstream is significantly higher that other land uses (Table 3) and there is significant correlation with runoff coefficient and soil erosion (Figure 6 and Figure 7 respectively) . Litter is part of the body of the plant (in the form of leaves, branches, twigs, flowers and fruit) that dies (deciduous or pruned) and lives on the surface of the soil either intact or partially weathered. The role of litter in maintaining infiltration and soil erosion through: (a) Maintaining soil looseness by protecting the soil surface from rainwater, so that aggregates and soil macro pores are maintained, (b) Providing food sources for soil organisms, especially 'soil diggers' (eg earthworms) ), so that the organism can live and develop in the soil. Thus the number of macro pores is maintained through the activity of these organisms, and (c) Maintaining water quality in the river through the filtering of soil particles carried by surface runoff before entering the river. In a study in North China, Li et al. (2014) showed that presence of litter of *Quercus variabilis*, representing broadleaf litter, and *Pinus tabulaeformis*, representing needle leaf litter, can reduce surface runoff rates by 29.5% and 31.3% respectively. The overall effect of fast plus slowly decomposing surface litter means protection of the soil surface from splash erosion, surface roughness that reduces sediment entrainment, an energy source for soil biota and a conducive microclimate (Hairiah et al. 2006, Derpsch et al., 2014).

The land surface roughness also determine to maintain high infiltration and reducing soil erosion. In upstream there is no significant different between land uses, but in midstream, land surface roughness in agroforestry systems with tightly different canopies is significantly higher than rare canopies (Table 3). Without a high canopy cover (Table 3.a), this roughness was not able to control surface runoff and erosion in the upstream area. This is due steep slope in this plot. Both the production forest and agroforestry systems with high canopy maintained a relatively high land surface roughness compared with rare canopies in midstream area..In midstream, the land surface roughness were significant correlations with runoff coefficient and soil erosion. The role of surface roughness as sediment filter may depend on frequent regeneration to counter homogenisation (Rodenburg et al. 2003). Surface roughness in the landscape includes a cavity, meandering of streams due to the present of litter, necromass, tree trunk and rocks, providing opportunities for water flow to stop for longer periods and experience infiltration. This condition also functions as a sediment filter. This function needs to be managed through land management, so that surface roughness is maintained on the ground.

The third research question is as an analysis the answers of the previous two research questions with the hypothesis that it is not always that the upstream watershed area is more sensitive to hydrological disturbance due to changes in land use than midstream, but the factor of soil properties also determines considerations in watershed hydrological management. From a land use policy perspective our results

suggest that maintaining high (~80%) canopy cover in the mid-slope farmer-controlled landscape that does not match the slope criteria for designation as watershed protection forest, is important. In Indonesia, protection is forest areas that have the primary function as protection of life support systems to regulate water management, prevent flooding, control soil erosion, prevent sea water intrusion, and maintain soil fertility (Government of Indonesia, 1999). With the higher rainfall intensities here and more erosive soils, risks for degradation from a downstream perspective seem to be as important here as they are in the more visually-at-risk upper watershed zone. Combining our plot-level results with efforts for hydrologic modelling for the Rejoso catchment as a whole (Tanika et al. 2018) can guide further advice to a local watershed forum on the measures and incentives needed to restore and protect the watershed as a whole. The Indonesian legal requirement of 30% forest cover across all its local government entities (Government of Indonesia, 2007) is a coarse translation of hydrologic relations at risk. It clearly matters what the land cover in the 'non-forest' parts of the landscape is and how vegetation interacts with soils and geomorphology in shaping rivers and groundwater flows (Zhipeng et al. 2018; Zhao et al. 2019). Our findings for the Rejoso watershed show that within the agroforestry spectrum, hydrologic thresholds of infiltration-friendliness exist between the systems that are mostly 'agro' and those that are mostly 'forest'.

Additional References:

Gifford, G.F., Hawkins, R.H.. Hydrologic impact of grazing on infiltration: a critical review. Water Resour. Res. 14, 305–313. 1978

Ma, X., Zhang, B., Shi, D., Lü, G.. Study on soil infiltration characteristic of different land utilization types in purple soil hilly region. J. Soil Water Conserv. 21, 25–29. 2007.

Neris, J., Jiménez, C., Fuentes, J., Morillas, G., Tejedor, M. Vegetation and land-use effects on soil properties and water infiltration of Andisols in Tenerife (Canary Islands, Spain). Catena 98, 55–62. 2012.

Suprayogo D., Widianto, Purnomosidi P., Widodo R.H., Rusiana, F, . Aini, Z.Z., Khasanah N. and Kusuma Z. Degradation of soil physical properties as as result of forest land use change to become a monoculture coffee system: study of soil macro-porosity degradation Agrivita 26 (1): 60-68. 2004.

Sun, D, ,Yang H. ,Guan D. ,Yang M., Wu J., Yuan, F., Wang A., Yushu Zhang Y., Jin C.. The effects of land use change on soil infiltration capacity in China: Ameta-analysis. Science of the Total Environment. 626:1394-1401. https://www.sciencedirect.com/science/article/pii/S0048969718301244?via%3Dihub, 2018.

Wang, L., Zhong, C., Gao, P., Xi, W., Zhang, S.. Soil infiltration characteristics in agroforestry systems and their relationships with the temporal distribution of rainfall on the Loess Plateau in China. PLoS One 10, e0124767. 2015

Widianto; Noveras, H.; Suprayogo, D.; Widodo, R.H.; Purnomosidhi, P. dan M. van Noordwijk. 2004. Conversion of Forests to Agricultural Land: Can the hydrological function of forests be replaced by monoculture coffee systems?. Agrivita 26 (1): 47-52. 2004.

Wu, Q.X., Chen, Y.M., Liu, X.D., Zhao, H.Y. Soil and Water Conservation Mechanism of Forest and it Function Control Technology. Chinese Science Press, Beijing, p. 118. 2005.

**Supplement Note 6:**

Table 3. Canopy cover, understory vegetation, litter necromass, and soil roughness of the sample plots

a. Upstream Rejoso watershed

| Code | Land cover | Tree canopy cover (%)* | Understorey vegetation (t ha$^{-1}$)* | Litter (t ha$^{-1}$)* | Soil roughness (%)* |
|---|---|---|---|---|---|
| UT1 | Old production forest | 55 b | 10.1 b | 9.2 b | 8.5 a |
| UT2 | Young pro-duction forest | 40 b | 10.5 b | 2.0 a | 7.0 a |
| UT3 | Agroforestry | 4 a | 10.1 b | 2.1 a | 9.5 a |
| UT4 | Arable land | 0 a | 3.7 a | 0.3 a | 7.7 a |
| LSD | | 15 | 5.6 | 3.7 | 4.6 |

b. Midstream Rejoso watershed

| Code | Land cover | Tree canopy cover (%)* | Understorey vegetation (t ha$^{-1}$)* | Litter (t ha$^{-1}$)* | Soil roughness (%)* |
|---|---|---|---|---|---|
| MT1 | Old production forest | 87 c | 2.5 a | 9.8 b | 7.6 b |
| MT2 | Agroforestry | 75 c | 2.5 a | 4.8 a | 5.4 ab |
| MT3 | Agroforestry | 52 b | 2.1 a | 5.2 a | 2.8 a |
| MT4 | Agroforestry | 26 a | 1.3 a | 3.5 a | 2.0 a |
| LSD | | 14 | 2.6 | 2.4 | 4.5 |

*The same letter indicates no statistically significant differences between location with Fisher's LSD test (p<0.05).*

Table 4. Rainfall, runoff, ratio runoff/rainfall and soil erosion in the runoff plots in each land cover type

a. Upstream Rejoso watershed

| Code | Land cover | Ranfall (mm) | Runoff (mm)* | Runoff/ rainfall ratio* | Soil erosion (ton ha$^{-1}$)* |
|---|---|---|---|---|---|
| UT1 | Old production forest | 555 | 14.3 a | 0.03 a | 5.86 a |
| UT2 | Young pro-duction forest | 492 | 13.2 a | 0.03 a | 1.47 a |
| UT3 | Agroforestry | 476 | 203.3 b | 0.43 b | 120.98 b |
| UT4 | Arable land | 556 | 225.7 b | 0.41 b | 163.22 b |
| LSD | | | 46.3 | 0.09 | 87 |

b. Midstream Rejoso watershed

| Code | Land cover | Ranfall (mm) | Runoff (mm)* | Runoff/ rainfall ratio* | Soil erosion (ton ha$^{-1}$)* |
|---|---|---|---|---|---|
| MT1 | Old production forest | 616 | 80.2 a | 0.13 a | 3.07 a |
| MT2 | Agroforestry | 841 | 316.3 c | 0.38 b | 2.88 a |
| MT3 | Agroforestry | 616 | 228.8 b | 0.37 b | 6.63 ab |
| MT4 | Agroforestry | 541 | 344.9 c | 0.64 c | 10.33 b |
| LSD | | | 86.6 | 0.12 | 4.22 |

*The same letter indicates no statistically significant differences between location with Fisher's LSD test (p<0.05).*